# A novel 3D bilateral filtering algorithm with noise level estimation assisted by multi-temporal SAR

**Haiyan Zhang, Yang Liu, Guoyin Cai***

School of Geomatics and Urban Spatial Informatic, Beijing University of Civil Engineering and Architecture, Beijing, China

* cgywj@126.com

**Data Availability Statement:** DOI:https://doi.org/10.6084/m9.figshare.27693714.v1.

**Funding:** The author(s) received no specific funding for this work.

## Abstract

The bilateral filter is widely employed in the field of image denoising due to its flexibility and efficiency. It calculates the weights of neighboring pixels based on both spatial and gray-scale distances from the pixel to be denoised. By incorporating the information of neighboring pixels through a weighted average, it reduces the disparity between the target pixel and its neighbors, achieving the goal of denoising. However, the extensive imaging range of SAR, coupled with low spatial resolution and the complexity of surface features, results in significant variations in the information expressed by each pixel within the kernel. Consequently, relying solely on neighboring pixel information for denoising can introduce a considerable amount of extraneous data into the target pixel, reducing image contrast and blurring edge contours. Additionally, because the noise levels in pixels of SAR images vary, the uniform filtering approach of the bilateral filter may lead to a degree of information loss in the filtered pixels. Ultimately, while the bilateral filter performs well in addressing additive noise, it is less effective against the multiplicative noise common in SAR images, further diminishing its filtering efficacy. To address these issues, we have developed the 3D bilateral filtering algorithm with noise level estimation assisted by multi-temporal SAR(3D-NLE-BF). This algorithm begins by evaluating the noise content of pixels to be denoised based on their temporal and spatial stability, classifying them into strong noise, weak noise, and noise-free pixels. Given the higher similarity of pixels along the temporal axis in multitemporal SAR data, the algorithm capitalizes on this feature to ensure that denoised pixels contain more useful information. Taking into account the characteristics of multitemporal SAR, the algorithm incorporates range-weight, spatial-weight, confidence-weight, and time-weight, designing corresponding filtering kernels for both strong and weak noise pixels. To verify the superiority of the algorithm, we selected Bilateral, NLM, Kuan, Lee, Lee-Enhanced, and Lee-Sigma as comparison algorithms. Real and simulated SAR denoising experiments were designed, and the denoising results were evaluated using ENL, SSI, PSNR, and QIUI, achieving favorable evaluation results. This demonstrated the effectiveness and general applicability of the algorithm proposed in this paper.

**Competing interests:** The authors have declared that no competing interests exist.

## 1. Introduction

Synthetic Aperture Radar (SAR) captures images of the earth's surface by emitting microwave signals and recording the reflected echoes. This technology allows SAR satellite imaging to function independently of daylight, providing continuous observation capabilities. Additionally, the long wavelengths of the microwaves enable penetration through clouds, ensuring reliable imaging regardless of weather conditions. SAR technology has been extensively applied in various fields, including forest detection [1, 2], vessel monitoring and classification [3–5], and urban surveillance [6]. However, the echo coherence phenomenon causes substantial coherent speckle noise in SAR images. As noted in [7], excessive noise reduces the contrast of SAR images, leading to significant loss of original information, which critically impacts the utility of the images. Therefore, effective denoising is essential before applying SAR images.

In recent years, SAR image denoising has emerged as a prominent research focus within SAR image preprocessing, where the primary challenge lies in effectively removing noise while retaining critical image details [8]. To address this, scholars have proposed various spatial domain denoising algorithms, which can be broadly classified into three main categories: methods based on local information, methods based on non-local information, and deep learning-based denoising methods.

Local methods, such as Lee [9], Kuan [10], and Frost [11], reduce noise based on pixel statistics within a neighborhood, using minimum mean square error or maximum a posteriori estimation. Due to the simplicity of the filtering rules in these methods, pixel information within the kernel is introduced indiscriminately into the pixels to be denoised. Consequently, the denoised pixels contain excessive irrelevant information, leading to blurred contours in the processed image. While effective, these methods often compromise image detail and texture [12]. Enhancements to Lee's algorithm [13] classify regions based on the coefficient of variation within the window, applying specific filters to homogeneous, heterogeneous, and noise-affected areas for improved results. Based on Frost and Lee [14], proposed a reconstructed decay factor, which was applied to SAR image processing, yielding more satisfactory filtering results.

Other local methods compute weights for neighboring pixels based on feature differences within the filtering window, averaging these weights for the final result. Gaussian filtering, a widely used weighted mean algorithm, assigns weights based on Euclidean distance. Bilateral filtering [15], an extension of Gaussian filtering, additionally considers grayscale differences, preserving detail and enhancing contrast. However, fixed spatial and range parameters can lead to suboptimal results, as they do not account for actual noise levels. An iterative bilateral filtering approach [16] adjusts these parameters based on noise levels and the coefficient of variation, enhancing denoising effectiveness. The ATS-RBF algorithm proposed in [17] utilizes the adaptive sample trimming approach to select suitable pixels within the reference window for filtering, effectively eliminating the influence of strongly noisy pixels on the filtering results. [18] combines local and non-local filtering techniques and employs the complex Wishart distance to measure pixel similarity in SAR images.

Images often exhibit self-similarity, leading to regions with similar characteristics [19]. Non-local methods, exemplified by the NLM algorithm [20], integrates information across the entire image. This approach constructs a reference block around the target pixel, computes weights based on similarity with matching blocks within a search area, and aggregates weighted pixel values for the final result. Since NLM relies on calculating the similarity between the pixel to be denoised and those within the search window, filtering complex images presents challenges. In such cases, the variation among pixels within the search window is often high, making it difficult to effectively integrate relevant pixel information into the target

pixel. This results in a diminished denoising effect. Lowering the similarity threshold to improve integration leads to greater loss of fine details in the denoised image, producing a more blurred visual outcome. Iteratively weighted PPB [21] addresses data redundancy in multi-temporal SAR images. The SAR-NLR-GBF algorithm [22] combines global and local statistical features, preprocessing guided images with an NLR-based filter for enhanced GBF effects. Inspired by NLM, the BM3D algorithm [23] exploits noise invariance under orthogonal transformation to separate low-frequency and high-frequency components, preserving detail and preventing edge blurring. Building on the BM3D algorithm, Sara and Chierchia introduced the SAR-BM3D [24] and MSAR-BM3D [25] algorithms specifically for denoising SAR images. The implementation of non-local filtering in SAR images has proven effective, resulting in the development of advanced despeckling techniques within a non-local framework, including FANS [26], NL-CV [27], and NL-SAR [28].

Deep learning has gained substantial traction across various domains [29, 30] by utilizing extensive training samples to model functional relationships between input and target outputs. In the area of SAR noise removal, however, the complexity and widespread distribution of SAR noise present challenges, leading to a shortage of optimal datasets for SAR denoising. To address this issue [31], created a SAR image denoising dataset by using the mean image of multi-temporal SAR as an approximation of a noise-free reference image. Most SAR denoising networks are primarily designed to address additive noise. To tackle the multiplicative nature of SAR noise [32], proposes SAR-DURNet, an approach that frames SAR denoising as an iterative process using half-quadratic splitting based on the a priori distribution of SAR noise. This approach enables noise removal through SAR-DURNet. Alternatively [33], introduces ID-CNN, which derives the noise component of the image using CNNs, then divides the original image by this noise component according to the multiplicative criterion to yield a denoised output.

In summary, most filters perform denoising solely based on the information contained in a single image, often leading to the introduction of excessive extraneous data, which results in blurred edge contours. Additionally, these methods typically do not account for the inherent noise levels present in the pixels, resulting in a loss of valuable information after denoising. To address these issues, we propose the 3D bilateral filtering algorithm with noise level estimation assisted by multi-temporal SAR(3D-NLE-BF), which classifies the pixels to be denoised into strong noise, weak noise, and noise-free categories based on their temporal and spatial stability. Furthermore, the algorithm is designed to incorporate range-weight, spatial-weight, confidence-weight, and time-weight based on the characteristics of multitemporal SAR. This approach leverages information from pixels located in the same spatial position but acquired at different times, effectively reducing the unnecessary data in the denoised pixels and enhancing the overall quality of the imagery. This approach achieves superior filtering effects and significantly enhances image readability. The main contributions of this paper are as follows:

1. The algorithm assesses the noise content in pixels targeted for filtering by integrating both their temporal and spatial stability and then categorizes them based on these evaluations. First, the coefficient of variation is calculated to assess temporal variation, yielding the temporal stability coefficient of the pixel to be denoised. Simultaneously, a central temporal phase pixel block is constructed around the target pixel, and its difference from a noise-free pixel block provides the spatial stability coefficient. Due to the absence of an actual noise-free block, this study uses the mean pixel block of the multi-temporal phase SAR as a substitute. Based on the temporal and spatial stability coefficients, pixels are then categorized into strong-noise, weak-noise, and noise-free types.

2. Filter weights aligned with multi-temporal SAR characteristics: To adapt the bilateral filter for SAR images, we retain the original spatial weight and redesign the range weight, as specified in Eq 5, to improve its handling of multiplicative noise. Additionally, since neighboring pixels are affected by noise to varying degrees, we introduce a confidence weight to mitigate the impact of pixels strongly influenced by noise on the filtering results. Finally, we design a time weight to reduce the effect of feature changes on filtering, further enhancing the overall quality of the results.

The remainder of the paper is organized as follows: Section II presents the current state of the work. Section III details the complete execution flow of the proposed 3D bilateral filtering algorithm with noise level estimation assisted by multi-temporal SAR. Section IV presents experimental results and analysis comparing this algorithm with other approaches. Finally, Section V summarizes the contributions and discusses future prospects.

## 2. Related works

### 2.1. SAR pixel similarity criterion

Since SAR image noise is multiplicative [34], the relationship between the observed intensity $I$, the speckle noise $S$, and the underlying backscattering coefficient $R$ in the image can be expressed by the following equation:

$$I = S \cdot R \tag{1}$$

The probability density function of $S$ follows the equation [35]:

$$P_S(S) = \frac{L^L S^{L-1} e^{-LS}}{\Gamma(L)} \tag{2}$$

Assuming that $R$ and $S$ are independent random variables, we can obtain:

$$P_{I|R}(I|R) = P_S\left(\frac{I}{R}\right)\frac{1}{R} = \frac{L^L I^{L-1}}{\Gamma(L)R^L}\exp\left(-L\frac{I}{R}\right) \tag{3}$$

where $L$ denotes the equivalent number of looks, and $\Gamma(\cdot)$ denotes the Gamma function.

Considering the multiplicative characteristics of SAR noise [36], employs Eq 4 to express the degree of similarity between two SAR image blocks, $h$ and $f$, of size $m \times n$. Respectively, $h(x, y)$ and $f(x,y)$ denote the grayscale values at the position coordinates $(x,y)$ in blocks $h$ and $f$.

$$S = \sum_{x=1}^{m}\sum_{y=1}^{n}\log\left[\sqrt{\frac{h(x,y)}{f(x,y)}} + \sqrt{\frac{f(x,y)}{h(x,y)}}\right] \tag{4}$$

When comparing only the degree of similarity between two pixels, Eq 4 simplifies to Eq 5.

$$S = \log\left[\sqrt{\frac{A_1}{A_2}} + \sqrt{\frac{A_2}{A_1}}\right] \tag{5}$$

### 2.2. Bilateral filter

The bilateral filter [15] comprises two components: spatial-weight and range-weight. It computes these weights based on the European distance and grayscale differences between the pixel to be filtered and neighboring pixels. This approach assigns lower weights to neighboring pixels with significant grayscale differences, thereby preserving image edges and details while achieving smoothing effects.

The original image $f$ and the image processed by the bilateral filter $h$. $f(x,y)$ and $h(x,y)$ respectively denote the pixel grayscale values at row $x$, column $y$ in $f$ and $h$.

$$h(x, y) = k^{-1}(x, y) \sum_{i=x-2}^{x+2} \sum_{j=y-2}^{y+2} f(i,j) w_d[f(x,y), f(i,j)] w_r[f(x,y), f(i,j)] \tag{6}$$

where $k(x,y)$ represents the regularization term, as illustrated in the following equation:

$$k(x, y) = \sum_{i=x-2}^{x+2} \sum_{j=y-2}^{y+2} w_d[f(x,y), f(i,j)] w_r[f(x,y), f(i,j)] \tag{7}$$

In the above equation, $w_d$ represents the spatial-weight, and $w_r$ represents the range-weight:

$$w_d[f(x,y), f(i,j)] = \exp\left[ -\frac{(x-i)^2 + (y-j)^2}{2(\sigma_d)^2} \right] \tag{8}$$

$$w_r[f(x,y), f(i,j)] = \exp\left[ -\frac{|f(x,y) - f(i,j))^2|}{2(\sigma_r)^2} \right] \tag{9}$$

## 3. Methods

The first part of this section outlines the execution flow of the algorithm, while the remainder provides detailed implementation specifics for each part of the algorithm.

### 3.1. Algorithmic process

The overall flow of the 3D-NLE-BF algorithm is shown below:
The detailed explanations of Fig 1 are shown below:

1. Input: The input of the algorithm in this paper is a SAR image sequence containing $2n+1$ scene images. The center time-phase image is the image to be denoised, while the $n$ scene images surrounding the center time-phase serve as auxiliary images for the filtering algorithm. At the same time, to minimize the impact of feature changes on the filtering algorithm, the acquisition times of multi-temporal SAR images should be as closely spaced as possible.

2. Pixel Classification Matrix Creation: First, the multi-temporal SAR is vectorized along the time axis to obtain a spatial vector matrix, in which each element is a spatial vector representing a set of pixels with the same spatial location and different acquisition times. Next, the temporal and spatial stability coefficients are calculated, and these are used to assess the noise content in the pixels to be denoised, resulting in a noise content coefficient. Finally, the noise content coefficient is processed using a classification threshold to determine the final pixel categories.

3. Pixels Denoising: The pixel type matrix categorizes the pixels in the image to be denoised into strong-noise pixels, weak-noise pixels, and noise-free pixels. In this process, the strong-noise pixels are first processed using 3D bilateral filtering, and the results are used to replace the strong-noise pixels in the image to obtain image $I_{n+1}'$. Next, the weak-noise pixels are processed using 2D improved bilateral filtering to produce image $I_{n+1}''$. Finally, since the noise-free pixels are undisturbed, image $I_{n+1}''$ becomes the final resultant image.

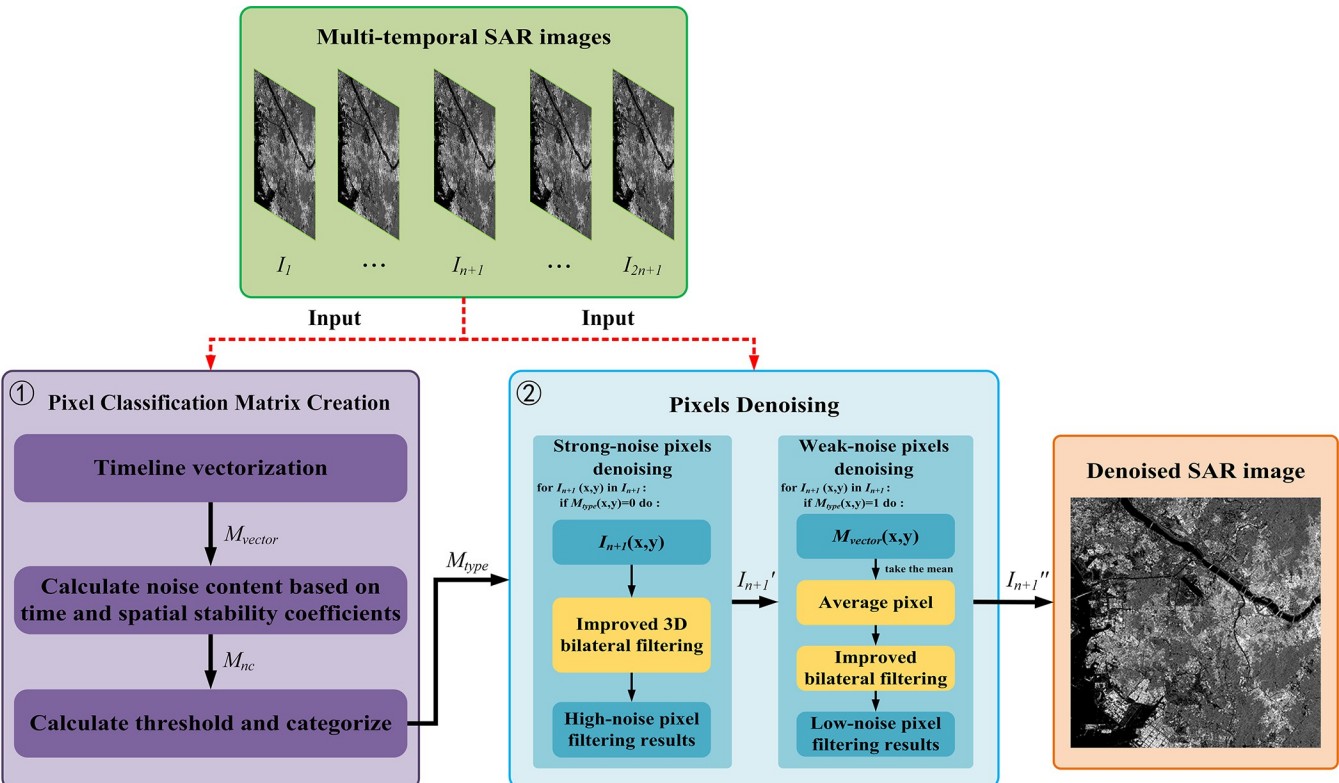

**Fig 1. Flowchart of the 3D-NLE-BF algorithm.**

## 3.2. Pixel classification matrix creation

**3.2.1. Spatial vector matrix creation.** There are $2n+1$ time series images with consistent shooting ranges, denoted by $I_1, \cdots, I_{n+1}, \cdots, I_{2n+1}$, where $I_{n+1}$ is the image to be denoised, and $I_1, \cdots, I_n, I_{n+2}, \cdots, I_{2n+1}$ are the denoised auxiliary images. Each image has $m$ rows and $n$ columns. Here, $I_t(x,y)$ represents the pixel with coordinates $(x,y)$ in the $t^{th}$ image of the image sequence. Pixels at the same spatial position along the time axis are assembled to obtain a spatial vector containing all the pixels at that spatial position. The construction process of the spatial vector is shown in Fig 2.

The expression of the spatial vector is shown in the following equation:

$$V(x, y) = [I_1(x, y), \cdots, I_{2n+1}(x, y)]^T \tag{10}$$

Then all the spatial vectors are organized according to their spatial positions to form the spatial vector matrix $M_{vector}$, which contains all the information of the multi-temporal SAR image, as illustrated in the equation below:

$$M_{vector} = \begin{bmatrix} V(1,1) & V(1,2) & \cdots & V(1,n) \\ V(2,1) & V(2,2) & \cdots & V(2,n) \\ \vdots & \vdots & \ddots & \vdots \\ V(m,1) & V(m,2) & \cdots & V(m,n) \end{bmatrix} \tag{11}$$

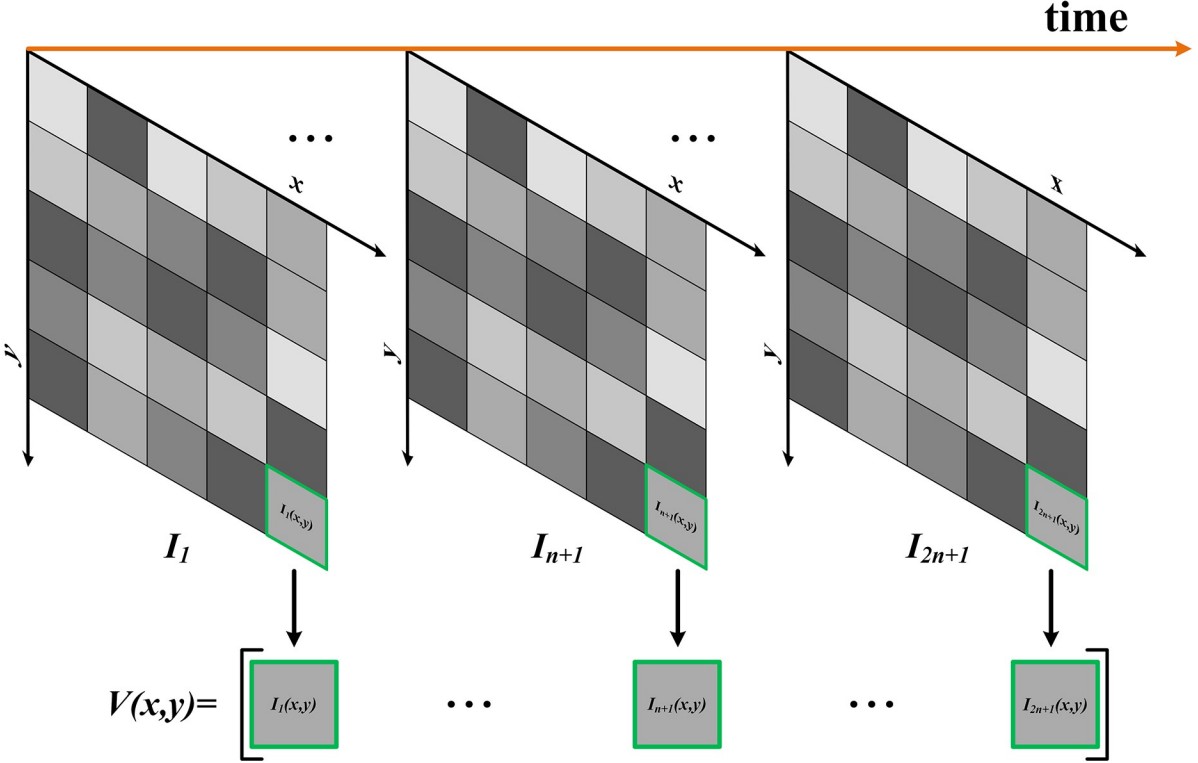

**Fig 2. The spatial vector organization schematic.**

**3.2.2. Noise content matrix creation.** The noise content matrix evaluates the noise content of the pixel to be denoised based on how stable it is in time and space.

*(1) Temporal stability coefficient.* The coefficient of variation is frequently employed to indicate the extent of data dispersion, neutralizing the influences of scale and magnitude. The coefficient of variation of the spatial vector reflects the degree of change in pixels along the time axis. Therefore, we use the coefficient of variation as the Temporal Stability Coefficient (TSC) to determine the temporal stability of the current pixel. The temporal stability coefficient $TSC(x,y)$ for spatial vector $V(x,y)$ can be calculated using the following equation:

$$TSC(x, y) = \frac{\sigma(x, y)}{\mu(x, y)} \tag{12}$$

where $\mu(x,y)$ and $\sigma(x,y)$ represent the mean and variance of the grayscale values of the pixels in the time vector $V(x,y)$, respectively.

$$\mu(x, y) = \frac{\sum_{t=1}^{2n+1} I_t(x, y)}{2n + 1} \tag{13}$$

$$\sigma(x, y) = \sqrt{\frac{\sum_{t=1}^{2n+1} [I_t(x, y) - \mu(x, y)]^2}{2n + 1}} \tag{14}$$

*(2) Spatial stability coefficient*. A larger temporal stability coefficient indicates that the pixel is less stable and more affected by noise. While the temporal stability coefficient provides a good measure of the pixel's stability over time, it only assesses the temporal variation of the pixel and does not consider the influence of spatially neighboring pixels, which results in an incomplete reflection of the pixel's noise content. To more accurately evaluate the noise content of the pixel to be denoised, we introduce the spatial stability coefficient.

Noise causes differences between the to-be-denoised pixels and noise-free pixels, and the relationship between them is depicted in Eq 1. The lower the noise content, the closer the ratio between the noisy pixels and noise-free pixels approaches 1. In order to calculate the spatial stability of the pixels to be denoised, we construct a central time pixel block centered on the pixel to be denoised and compare the similarity with the noise-free pixel block. Since noise-free pixel blocks are unavailable, we draw inspiration from multi-view processing and substitute the mean pixel blocks of multi-temporal SAR for the noise-free pixel blocks. A smaller spatial stability coefficient corresponds to lower pixel noise content. The calculation formula is as follows:

$$SSC(x,y) = \frac{\sum_{i=x-n}^{x+n} \sum_{j=y-n}^{y+n} ratio(i,j)}{(2n+1)^2} \tag{15}$$

In the above equation, $n$ is the radius of the pixel block, which is generally taken as 1. *ratio* (•) denotes the ratio function, which is calculated as follows:

$$ratio(x,y) = |1 - \frac{f(x,y)}{f_{mean}(x,y)}| \tag{16}$$

*(3) Noise content matrix*. The noise content matrix is composed of noise content coefficients, which are derived from the temporal and spatial stability coefficients. These coefficients account for the degree of stability of the pixel both in time and space, as shown below:

$$NCC(x,y) = \exp[TSC(x,y) + SSC(x,y)] - 1 \tag{17}$$

The noise content matrix is obtained by arranging the noise content coefficients by spatial location as shown below:

$$M_{nc} = \begin{bmatrix} NCC(1,1) & NCC(1,2) & \cdots & NCC(1,n) \\ NCC(2,1) & NCC(2,2) & \cdots & NCC(2,n) \\ \vdots & \vdots & \ddots & \vdots \\ NCC(m,1) & NCC(m,2) & \cdots & NCC(m,n) \end{bmatrix} \tag{18}$$

**3.2.3. Pixel classification matrix creation.** The pixels in the image to be denoised are categorized as follows: strong-noise pixels, weak-noise pixels, and noise-free pixels. Given that SAR images typically exhibit significant noise, this paper assumes that a majority of pixels are heavily affected by noise, categorizing them as strong-noise pixels, which generally constitute 95% of the total pixel count in the image. Weak-noise pixels are less affected by noise, and noise-free pixels are unaffected, together accounting for 5% of the total pixel count.

According to this rule, the elements in the noise content matrix are arranged in ascending order from smallest to largest. The $[(m \times n)/20]^{th}$ noise content value is then taken as the threshold $threshold_{nc}$. Pixels to be filtered are categorized by the following rule:

$$type(x, y) = \begin{cases} 0 & M_{nc}(x, y) > threshold_{nc} \\ 1 & 0 < M_{nc}(x, y) \leq threshold_{nc} \\ 2 & M_{nc}(x, y) = 0 \end{cases} \tag{19}$$

If $type(x,y) = 0$, then the pixel $I_{n+1}(x,y)$ to be filtered is classified as a strong-noise pixel, indicating it is heavily affected by noise. If $type(x,y) = 1$, the pixel $I_{n+1}(x,y)$ is categorized as a weak-noise pixel, implying it is less affected by noise. If $type(x,y) = 2$, the pixel $I_{n+1}(x,y)$ is considered a noise-free pixel, unaffected by noise. The final arrangement of $type(x,y)$ by spatial location results in the pixel classification matrix $M_{type}$, presented in the following form:

$$M_{type} = \begin{bmatrix} 0 & 0 & \cdots & 1 \\ 1 & 2 & \cdots & 0 \\ \vdots & \vdots & \ddots & \vdots \\ 1 & 2 & \cdots & 1 \end{bmatrix} \tag{20}$$

## 3.3. Pixels denoising

This section describes the composition of range-weight, spatial-weight, confidence-weight, and time-weight in detail. It also outlines the design of the filter kernel for addressing strong-noise pixels and weak-noise pixels.

**3.3.1. Constructing filter weights.** In this paper, we retain the spatial-weight from bilateral filtering while redesigning the range-weight. Additionally, we introduce confidence-weight and time-weight, incorporating features specific to multi-temporal SAR images. The detailed composition of each weight is provided below.

*(1) The range-weight.* Since the kernel filter relies on neighboring pixel information for denoising, neighboring pixels with significant grayscale differences from the pixel to be denoised can heavily influence the result, introducing unnecessary information that leads to the loss of the original pixel data. To mitigate this, the traditional bilateral filter uses range-weighting to reduce the impact of neighboring pixels with large grayscale differences on the denoising outcome. The range-weight employs Euclidean distance to measure the grayscale difference between neighboring pixels and the pixel to be denoised, which works well for Gaussian additive noise. However, since SAR noise is multiplicative, the range-weight needs to be adapted to more effectively measure the grayscale differences between SAR pixels.

The SAR pixel similarity criterion, as described in Eq 5, effectively captures the degree of similarity between SAR pixels. Therefore, the range-weight is redefined based on Eq 5 to measure the similarity between SAR pixels. The greater the grayscale difference between two pixels, the smaller the range-weight becomes. The improved range-weight $g(i,j,k)$ is obtained through Eq 21.

$$g(i, j, k) = \lambda_g \cdot norm[SIM(i, j, k)] \tag{21}$$

$$SIM(i, j, k) = \exp\left\{ \log\left[ \sqrt{\frac{f(i, j, k)}{f(x, y, z)}} + \sqrt{\frac{f(x, y, z)}{f(i, j, k)}} \right] \right\} \tag{22}$$

In the above equation: $norm(\bullet)$ represents the normalization function. $\lambda_g$ is a control parameter used to adjust the proportion of the range-weight in the overall filtering kernel weight. The larger the value of $\lambda_g$, the more significant the role of the grayscale difference in the filtering process.

*(2) The spatial-weight*. Due to the self-similarity of images, pixels that are closer to the target pixel for filtering tend to share more similar information. To mitigate the impact of distant pixels on the filtering result, we introduce a spatial-weight. This spatial-weight is calculated using a standard two-dimensional Gaussian function, optimizing the retention of original information in the filtered pixels. The spatial-weight $d(i,j,k)$ can be expressed as:

$$d(i, j, k) = \lambda_d \cdot norm[DIS(i, j, k)] \tag{23}$$

$$DIS(i, j, k) = \frac{1}{2\pi} \exp\left[-\frac{(i - x)^2 + (j - y)^2}{2}\right] \tag{24}$$

In the above equation: $norm(\bullet)$ represents the normalization function. $\lambda_d$ is a control parameter that adjusts the proportion of the spatial-weight in the overall filtering kernel weight. The bigger the value of $\lambda_d$, the more information is preserved in the pixels to be filtered.

*(3) The confidence-weight*. Due to varying noise contents across pixels, those with high noise levels experience significant information loss and cannot provide reliable feature information for pixels with low confidence. To mitigate the impact of high-noise pixels on the denoising result, this paper introduces a confidence-weight. This weight reduces the influence of high-noise pixels during the denoising process, ensuring that the denoised pixels retain more authentic information from their neighboring pixels.

Confidence-weight $p(i,j,k)$ is similar to the calculation of noise content coefficient, which evaluates the noise content of pixels according to the spatial stability coefficient and temporal stability coefficient, as follows:

$$p(i, j, k) = \lambda_p \cdot norm[CONF(i, j, k)] \tag{25}$$

$$CONF(i, j, k) = \exp[TSC(i, j) + SSC(i, j, k)] \tag{26}$$

In the above equation: $norm(\bullet)$ represents the normalization function. $TSC(\bullet)$ and $SSC(\bullet)$ represent the temporal and spatial stability coefficients, respectively, and are calculated as shown in Eq 12 and Eq 15. $\lambda_p$ is a control parameter used to adjust the proportion of the confidence-weight in the overall weighting of the filtering kernel. A larger value of $\lambda_p$ increases the influence of the confidence-weight in the filtering process.

*(4) The time-weight*. Coltuc proposed a multi-temporal SAR denoising method [37], which categorized the ground surface change factor into the low-frequency part after applying the DCT transform to multi-temporal SAR sequences. When the time interval is small, the impact of ground changes on multi-temporal SAR denoising is minimal. However, this paper argues that with longer time intervals in multi-temporal SAR, changes in ground features can influence denoising. To mitigate this effect, the time-weight reduces the weight of the filter kernel for images with larger time intervals. The time-weight $t(i,j,k)$ can be expressed as:

$$t(i, j, k) = \lambda_t \cdot norm\{\exp[-|t(i, j, k) - t(x, y, z)|]\} \tag{27}$$

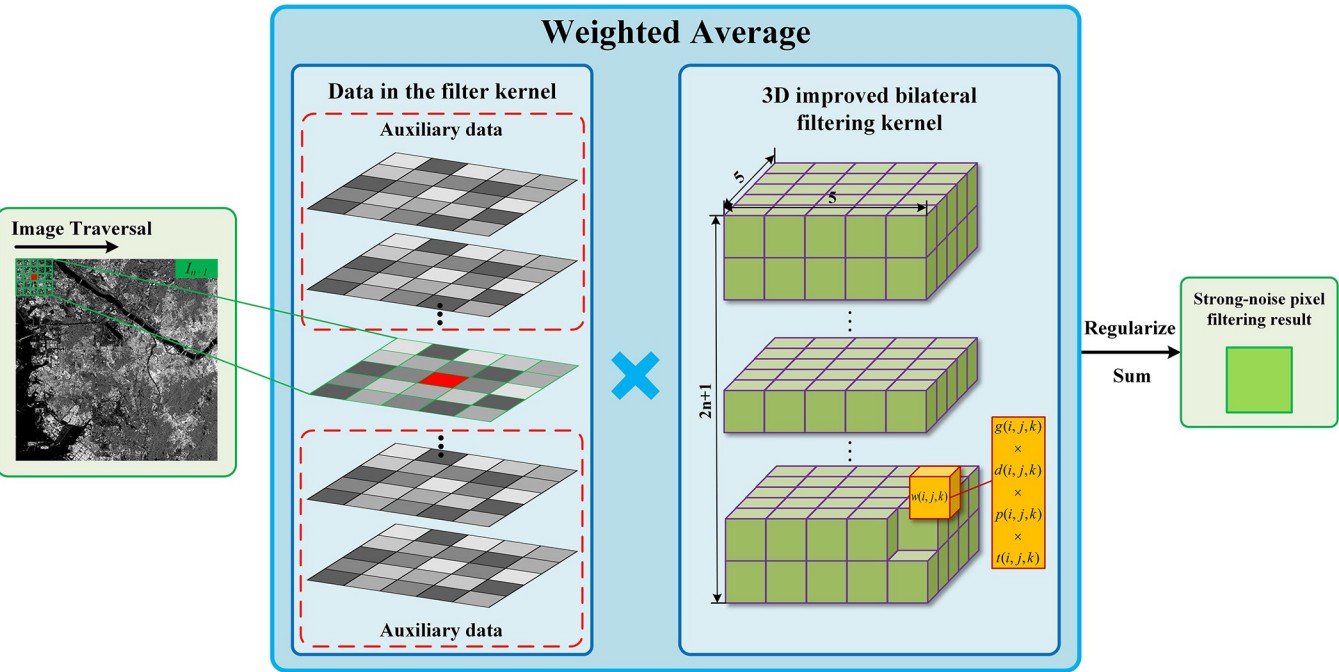

**Fig 3. Strong-noise pixel denoising process schematic.**

In the above equation: $norm(\bullet)$ represents the normalization function. $\lambda_t$ is a control parameter that adjusts the proportion of the confidence variance coefficient in the overall filtering kernel weight. A larger value of $\lambda_t$ increases the influence of the time variance coefficient in the filtering process. $t(\cdot)$ denotes the pixel acquisition time in days.

**3.3.2. Strong-noise pixel denoising.** Since strong-noise pixel occupy the majority of the image, their processing directly affects the quality of the resultant image. Therefore, we first filter the strong-noise pixel using the 3D improved bilateral filter kernel to obtain the intermediate result image $I_{n+1}{}'$. Fig 3 illustrates the filtering flow of the strong-noise pixel:

When the spatial range of the filter kernel is too large, it introduces excessive irrelevant information. Therefore, in this paper, the length and width of the 3D improved bilateral filter kernel are fixed to 5×5 respectively. Meanwhile, to comprehensively utilize all the information from SAR time series images, the height of the filter kernel is set to $2n+1$.

Considering the differences among the pixels within the filter kernel, this paper comprehensively calculates the weights based on four-dimensional differences: grayscale difference, confidence difference, acquisition time difference, and spatial distance difference. The weights can be expressed by the following equation:

$$w(i, j, k) = \exp[g(i, j, k) + d(i, j, k) + p(i, j, k) + t(i, j, k)] \tag{28}$$

In the above equation, $(x,y,z)$ denote the coordinates of the pixels to be filtered, and $(i,j,k)$ denote the coordinates of the neighboring pixels within the filter kernel. The weights of the neighboring pixels are represented by $w(i,j,k)$. The weights $g(i,j,k)$, $d(i,j,k)$, $p(i,j,k)$, $t(i,j,k)$ respectively denote the four components that constitute the pixel weights: the range-weight, the spatial-weight, the confidence-weight, and the time-confidence.

The relationship between the original pixel and the denoised pixel can be expressed by the following equation:

$$h(x, y, z) = \frac{\sum_{i=x-2}^{x+2} \sum_{j=y-2}^{y+2} \sum_{k=1}^{2n+1} w(i,j,k) f(i,j,k)}{\sum_{i=x-2}^{x+2} \sum_{j=y-2}^{y+2} \sum_{k=1}^{2n+1} w(i,j,k)} \tag{29}$$

**3.3.3. Weak-noise pixel denoising.** The weak-noise pixels exhibit only a minor degree of noise. Given that these pixels within spatial vectors share identical surface features, they demonstrate the highest self-similarity compared to the neighboring pixels. Therefore, denoising weak-noise pixels should heavily utilize the information from their respective spatial vectors. In this study, to leverage multi-temporal SAR advantages for denoising weak-noise pixels, we first calculate the mean value of the spatial vectors they belong to, yielding a mean pixel that consolidates multi-temporal information. Subsequently, this mean pixel undergoes 2D improved bilateral filtering to achieve the denoising result for the weak-noise pixel. The denoising process for weak-noisy pixels is shown in Fig 4:

The two-dimensional improved bilateral filter is derived from the three-dimensional improved bilateral filter and comprises two components: the range-weight and the spatial-weight. The weights $w(i,j)$ can be calculated using the following equation:

$$w(i,j) = \exp[g(i,j) + d(i,j)] \tag{30}$$

$g(i,j)$ and $d(i,j)$ denote the range-weight and spatial-weight of the neighboring pixels with coordinates $(i,j)$, respectively.

The relationship between the weak-noise pixel $f(x,y)$ and the denoised pixel $h(x,y)$ is shown in the following equation:

$$h(x, y) = \frac{\sum_{i=x-2}^{x+2} \sum_{j=y-2}^{y+2} w(i,j) f(i,j)}{\sum_{i=x-2}^{x+2} \sum_{j=y-2}^{y+2} w(i,j)} \tag{31}$$

## 3.4. Parameter setting

Based on extensive experiments, this paper recommends setting the parameters $\lambda_g, \lambda_d, \lambda_t, \lambda_p$ to 1.5, 3, 1 and 3, respectively, to achieve optimal filtering effects. Depending on the specific usage scenario, the $\lambda_g$ can be increased to preserve more image details. To achieve a smoother image, the $\lambda_d$ can be lowered. If retaining more original information in the filtered image is desired, the $\lambda_t$ can be raised. To reduce the impact of unstable pixels on the filtered image, increase the $\lambda_p$.

## 4. Experiment and results

In this section, to evaluate the effectiveness of 3D-NLE-BF, we select Bilateral, NLM, Kuan, Lee, Lee-Enhanced, and Lee-Sigma as comparison algorithms for the denoising experiments, with the parameter configurations of each algorithm provided in Table 1. The first part of this section presents comparison experiments using real SAR images. Additionally, since certain performance metrics require noise-free images as a reference, we conduct denoising experiments with simulated SAR images in the second part. The final part of this section compares the execution times of each algorithm to assess their time complexity.

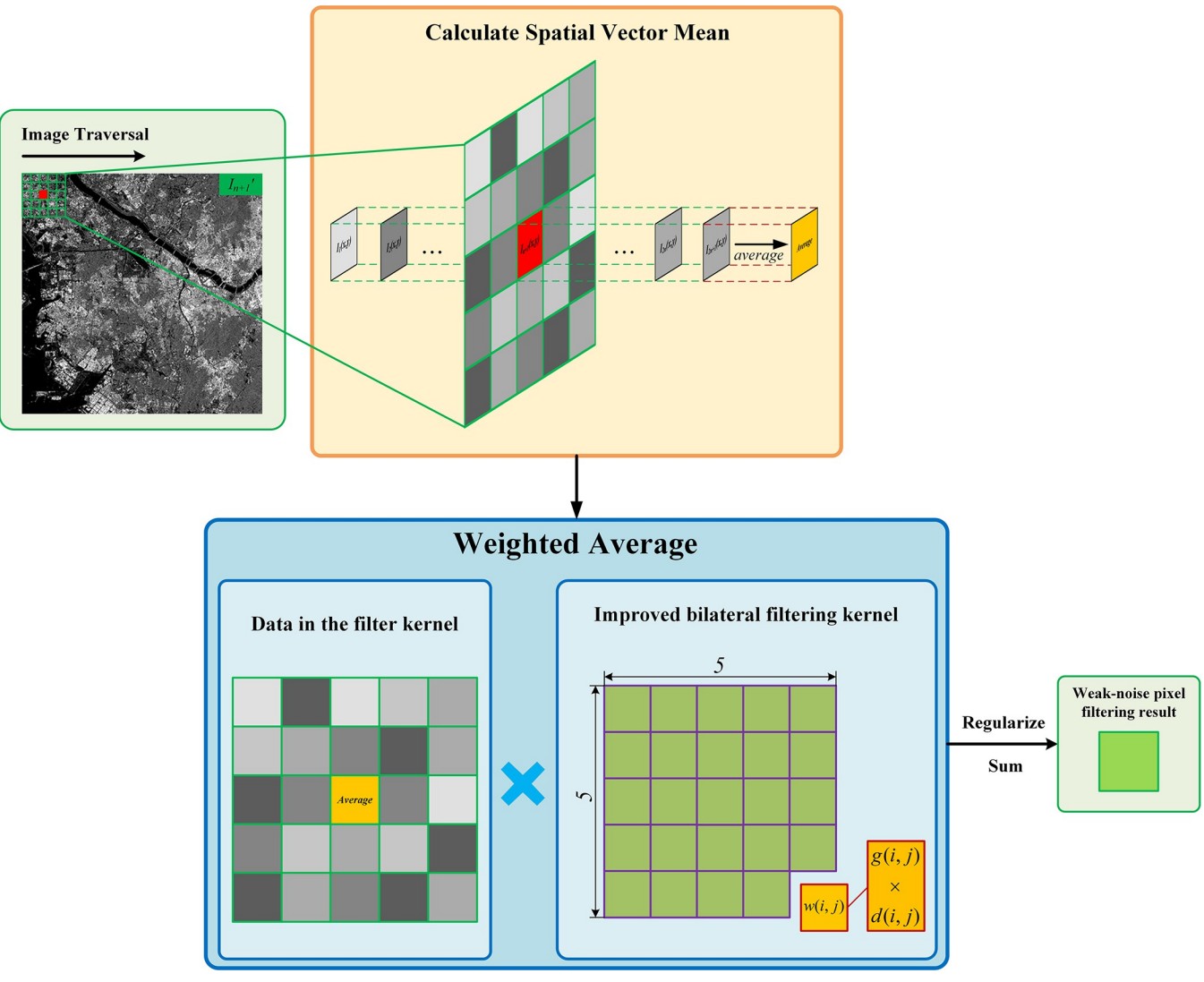

**Fig 4. Weak-noise pixel denoising process schematic.**

**Table 1. Algorithm parameter setting.**

| Algorithm | Algorithmic parameter |
|---|---|
| Bilateral | $WindowSize = 5, SigmaColor = 75, SigmaSpace = 75$ |
| NLM | $WindowSize = 5, SearchWindowSize = 11, Sigma = 10$ |
| Kuan | $WindowSize = 5, CU = 0.25$ |
| Lee | $WindowSize = 5, CU = 0.25$ |
| Lee-Enhanced | $WindowSize = 5, CU = 0.535, K = 1.0, CMAX = 1.73$ |
| Lee-Sigma | $WindowSize = 5, Sigma = 0.9, TK = 5, Looks = 1$ |
| 3D-NLE-BF | $WindowSize = 5, \lambda_d = 2, \lambda_t = 1, \lambda_g = 1.5, \lambda_c = 3$ |

## 4.1. Real SAR denoising experiment

**4.1.1. Experiment data.** In this experiment, VH polarization mode images captured by the Sentinel-1A satellite are selected for filtering. Each image is 500×500 pixels in size, with 5-looks. Additionally, four experimental regions are established: urban, coastal, farmland, and mountain. The data used in this experiment were obtained for free from Copernicus Open Access Hub (https://scihub.copernicus.eu/). Fig 5 shows the four regions to be processed, with yellow and blue labeled regions indicating homogeneous areas used to evaluate each algorithm's processing capability in these regions.

The first region is called urban (Fig 5(A)). The image was taken on April 25, 2024, and the auxiliary images entered into the algorithm were taken on March 20, 2024, April 1, 2024, April 13, 2024, May 7, 2024, May 19, 2024, and May 31, 2024. This region includes key city components such as rivers, roads, housing areas, and some farmland on the city's outskirts. It

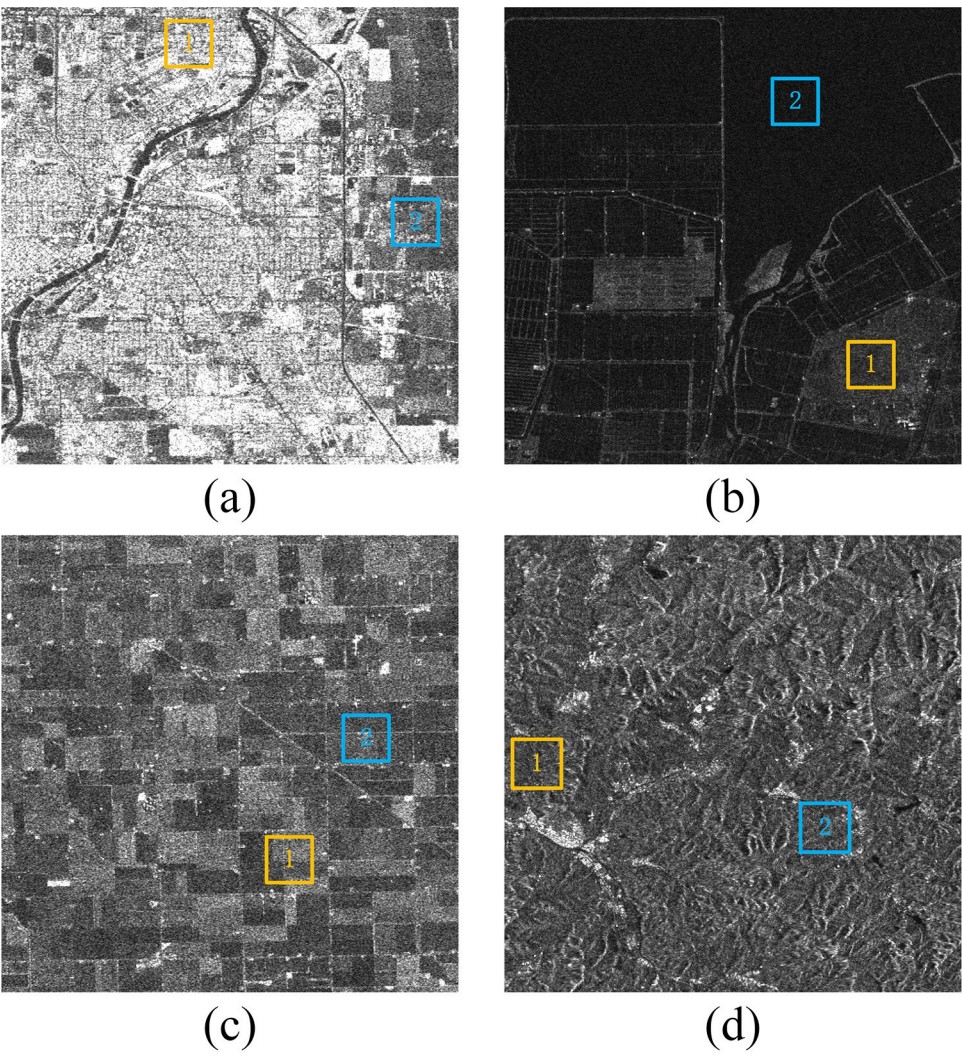

(a)

(b)

(c)

(d)

**Fig 5. Four images need to be denoised, with yellow and blue marked areas indicating homogeneous regions.** (a) urban. (b) coastal. (c) farmland. (d) mountain. Sentinel imagery was freely downloaded from the Copernicus Open Access Hub (https://scihub.copernicus.eu/).

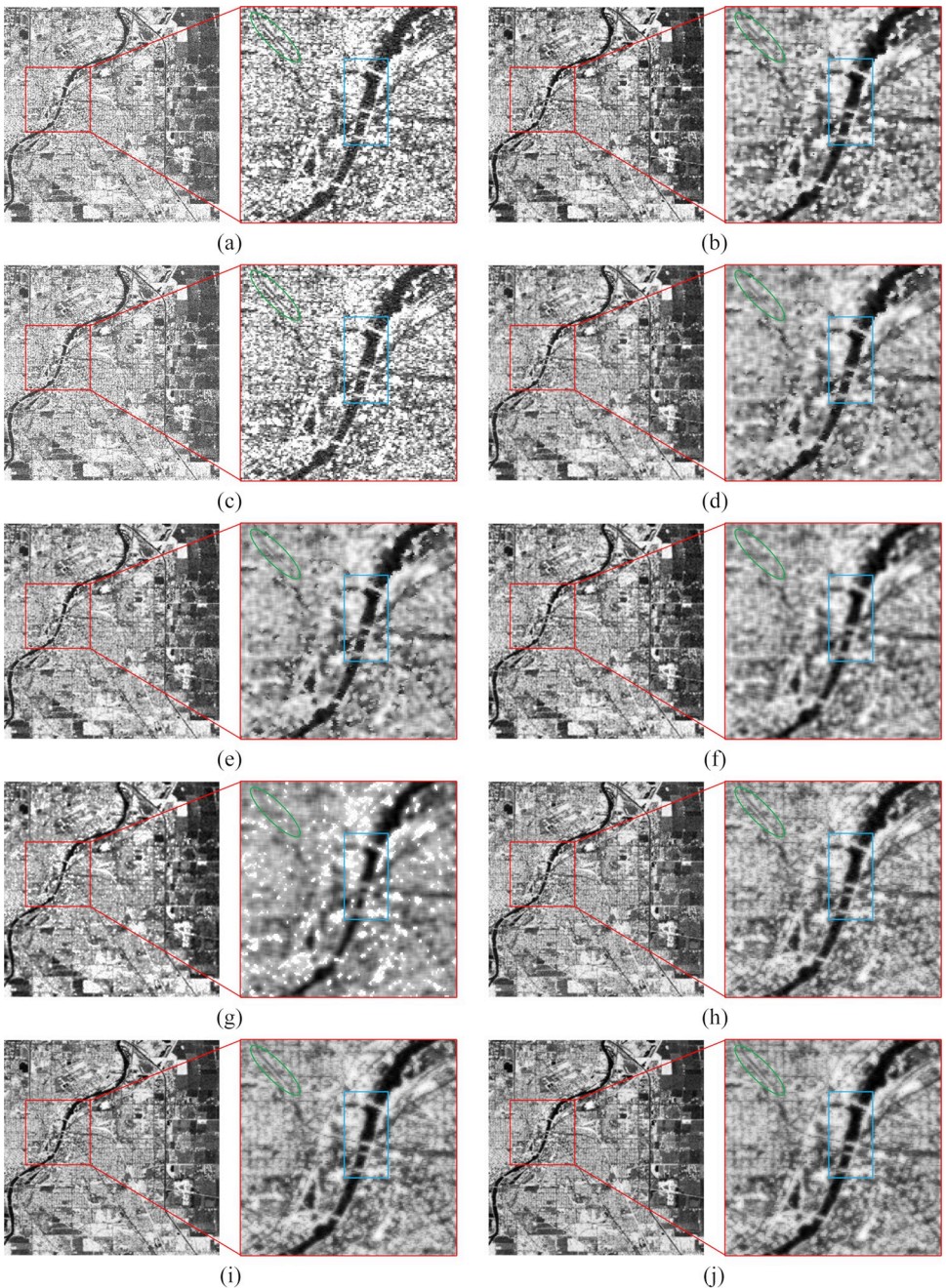

**Fig 6. The denoising results of the urban region.** (a) Original image. (b) Bilateral. (c) NLM. (d) Kuan. (e) Lee. (f) Lee-Enhanced. (g) Lee-Sigma. (h-j) 3D-NLE-BF with inputs of 2, 4, and 6 auxiliary images.

contains numerous strong feature points and is used to test each algorithm's ability to retain original information, with results shown in Fig 7.

The second region, named coastal, is a seafood artificial farm (Fig 5(B)). The image was taken on October 18, 2023, and the auxiliary images entered into the algorithm were taken on September 12, 2023, September 24, 2023, October 6, 2023, November 11, 2023, November 23, 2023, and January 10, 2024, respectively. This region contains substantial edge information,

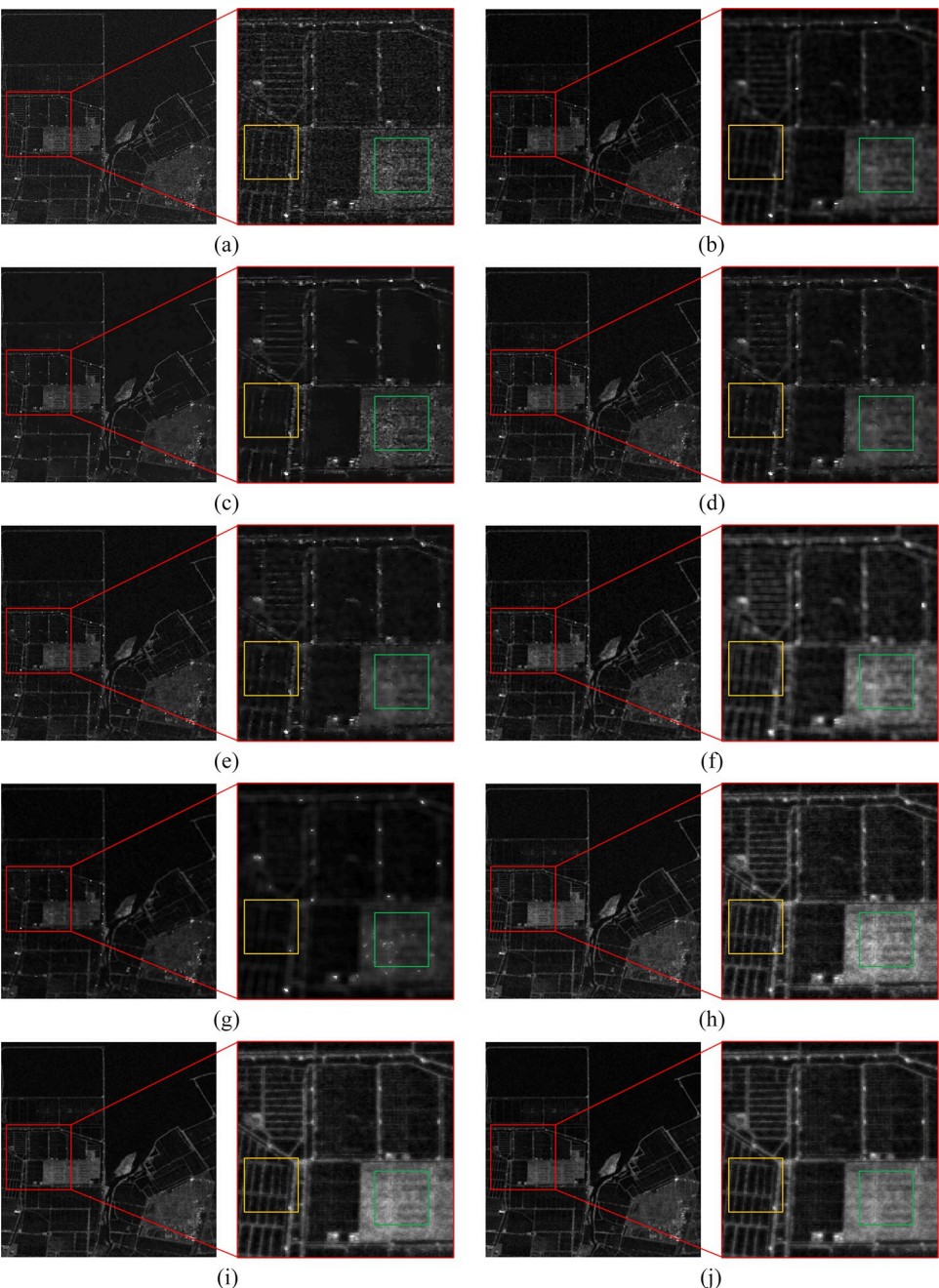

**Fig 7. The denoising results of the coastal region.** (a) Original image. (b) Bilateral. (c) NLM. (d) Kuan. (e) Lee. (f) Lee-Enhanced. (g) Lee-Sigma. (h-j) 3D-NLE-BF with inputs of 2, 4, and 6 auxiliary images.

making it suitable for testing the edge retention ability of each algorithm, with results shown in Fig 8.

The third region is farmland (Fig 5(C)), consisting of block fields with regular shapes and grayscale differences between blocks. The image was taken on April 25, 2024, and the auxiliary images entered into the algorithm were taken on March 20, 2024, April 1, 2024, April 13, 2024, May 7, 2024, May 19, 2024, and May 31, 2024. This region is primarily used to test the image contrast and contour shape distortion after algorithm processing, with results shown in Fig 9.

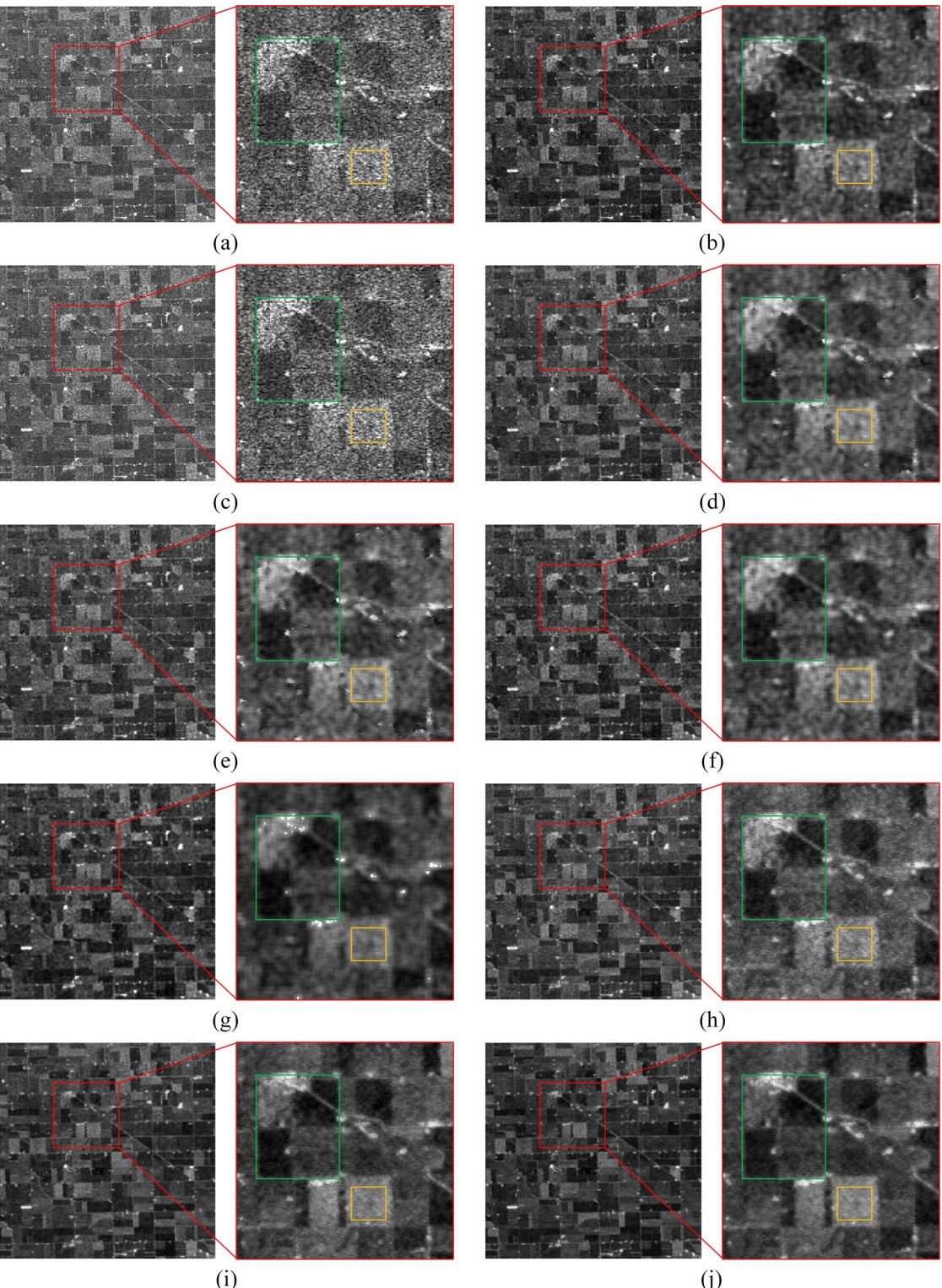

**Fig 8. The denoising results of the farmland region.** (a) Original image. (b) Bilateral. (c) NLM. (d) Kuan. (e) Lee. (f) Lee-Enhanced. (g) Lee-Sigma. (h-j) 3D-NLE-BF with inputs of 2, 4, and 6 auxiliary images.

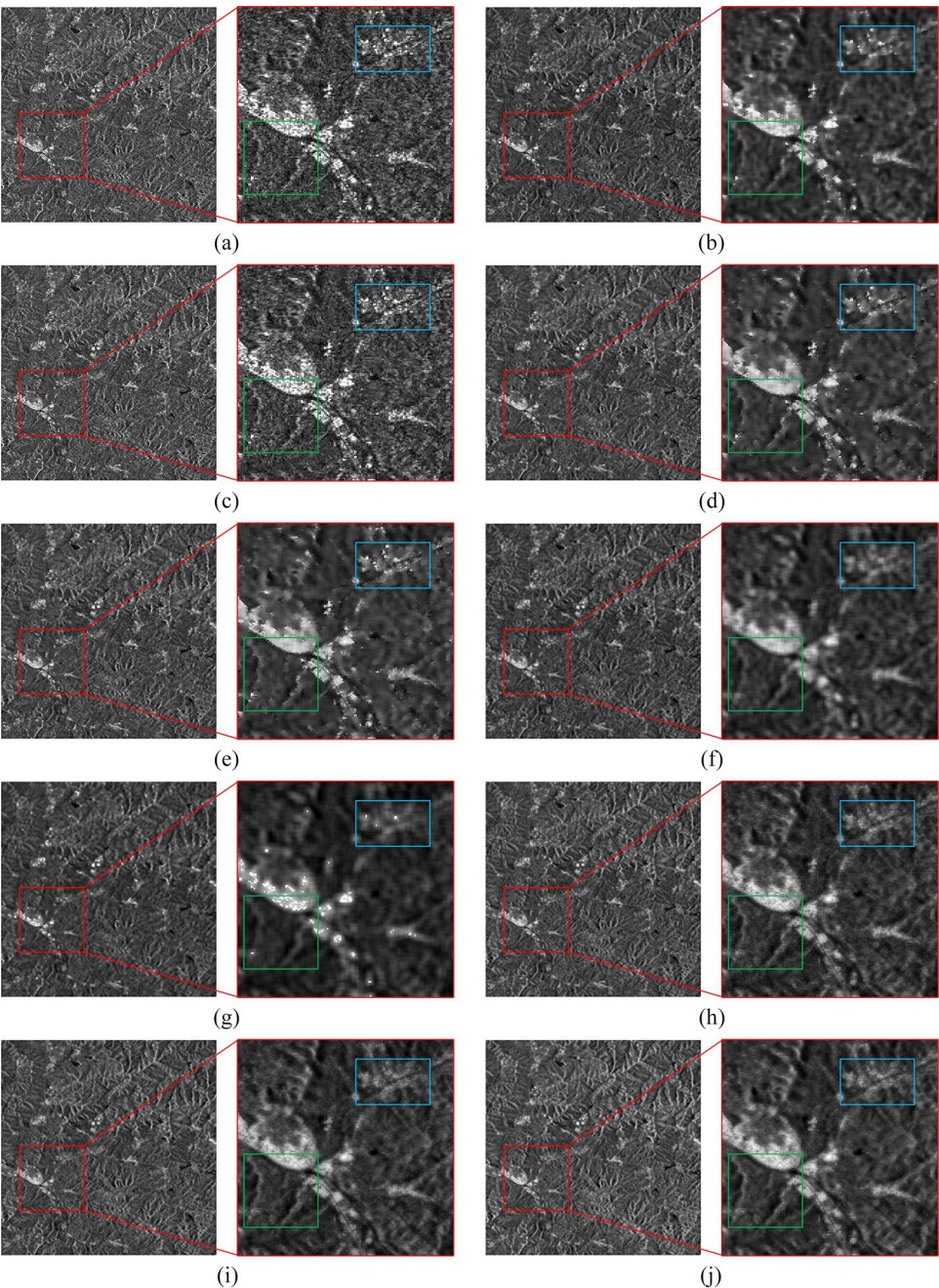

**Fig 9. The denoising results of the mountain region.** (a) Original image. (b) Bilateral. (c) NLM. (d) Kuan. (e) Lee. (f) Lee-Enhanced. (g) Lee-Sigma. (h-j) 3D-NLE-BF with inputs of 2, 4, and 6 auxiliary images.

The fourth region is mountain (Fig 5(D)), with a complex pixel distribution containing classic mountain components. The image was taken on January 20, 2024, and the auxiliary images entered into the algorithm were taken on December 15, 2023, December 27, 2023, January 8, 2024, February 1, 2024, February 13, 2024, and February 25, 2024, respectively. This region tests the algorithms' ability to maintain details, with processing results shown in Fig 10.

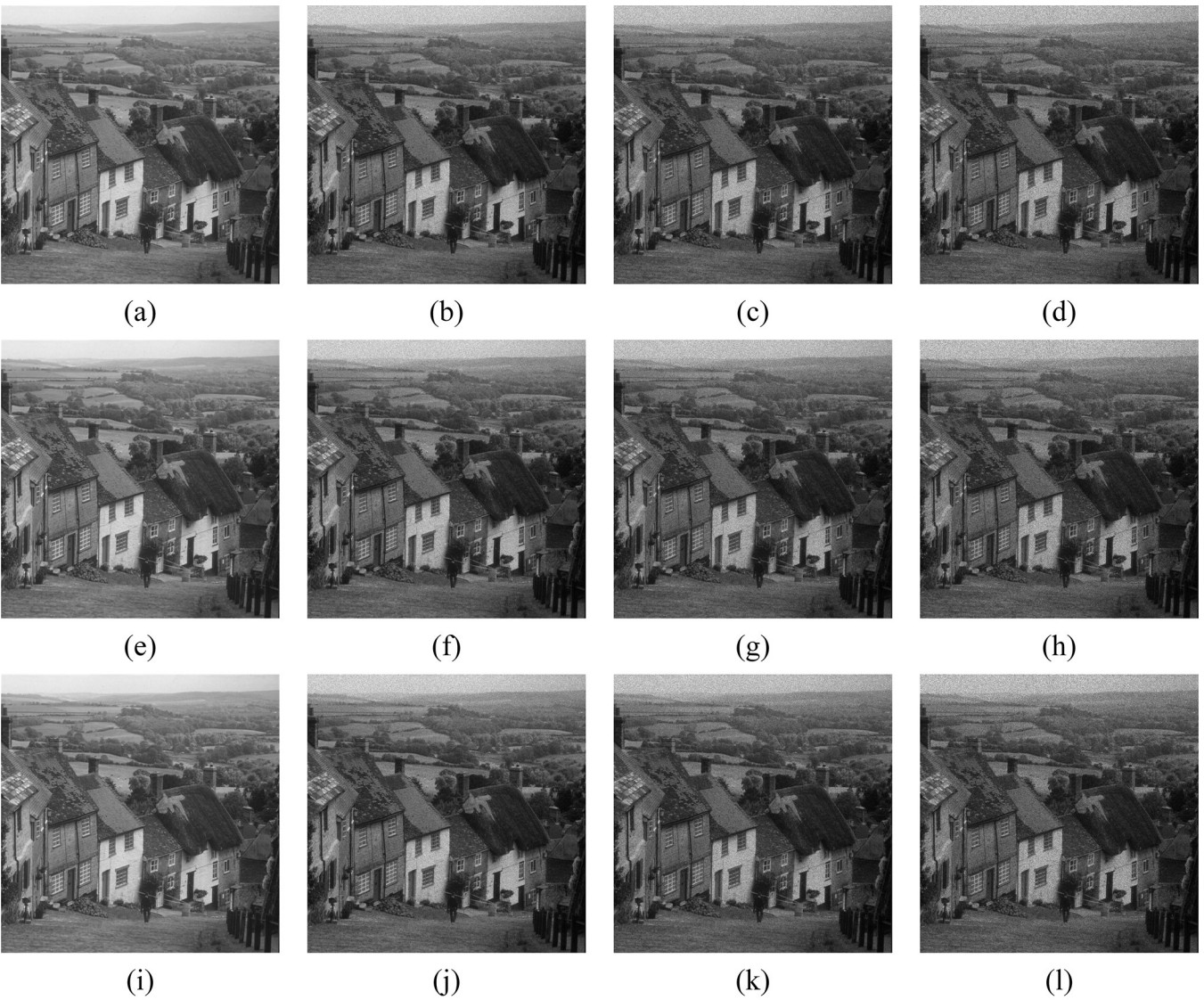

**Fig 10. Simulated SAR image.** (a) Goldhill. (b-d) Goldhill with noise, and the noise variance is 0.06, 0.08, and 0.1 in that order. (e) Lena. (f-h) Lena with noise, and the noise variance is 0.06, 0.08, and 0.1 in that order. (i) Peppers. (j-l) Peppers with noise, and the noise variance is 0.06, 0.08, and 0.1 in that order.

#### 4.1.2. Results and analysis.

The denoising results of the real SAR images are presented in Figs 6–9, where we provide a detailed analysis of the results for each algorithm, highlighting their respective advantages and disadvantages.

Bilateral: Due to the inability of bilateral filtering to adaptively adjust the sizes of the range-weight and spatial-weight, the contours of the filtered image become irregular. As shown in Figs 6–9(B), bilateral filtering significantly suppresses noise in the image compared to the original. In the green region of Fig 6(B), the yellow area of Fig 7(B), and the green section of Fig 9(B), some image details are preserved. However, as seen in the blue region of Fig 6(B), the yellow area of Fig 7(B), and the green region of Fig 8(B), the contours of the features become more distorted and do not align with the true feature characteristics.

NLM: Since NLM extracts information from the global to filter, the filtering effect is better in simpler images. As shown in Fig 7(C), NLM can effectively suppress the noise on the sea

surface, and the details of the yellow area are also preserved. However, as shown in Figs 6–9 (C), due to the small number of similar regions in the image, the noise of the image after NLM processing is not effectively suppressed, so NLM cannot process the complex image with large differences in grayscale and more details.

Kuan: This filter is primarily designed for additive noise, while SAR noise is multiplicative in nature, resulting in some degree of detail loss in the denoised image. From the green area in Fig 6(D), the yellow area in Fig 7(D), the green area in Fig 8(D), and the green area in Fig 9 (D), it can be seen that the Kuan filter can retain the details of the image better. However, as shown in the sea surface of Fig 7(D) and the yellow area of Fig 9(D), there is a large difference in the gray scale of the homogeneous part of the image, so the Kuan filter has poorer processing ability for the homogeneous area.

Lee: It performs denoising based on local statistics but does not take into account the differences between pixels within the filter's kernel range, leading to blurred edges in the denoised image. As shown in the green area of Fig 6(E) and the yellow area of Fig 7(E), the Lee filter has certain detail retention ability and better homogeneous area processing ability. However, the same phenomenon of blurred edges and irregular contours exists.

Lee-Enhanced: This filter, building upon Lee's method, introduces a weighted coefficient that can be dynamically adjusted based on the noise level. However, since the filtering result is influenced by the pixels within the kernel, significant grayscale differences between these pixels can lead to blurred edges. From Fig 6(F) blue region, Fig 7(F) yellow region, Fig 8(F) green region, it can be seen that Lee-Enhanced filter has better processing effect on the edge, and the contour is regular, which is in line with the feature characteristics. And from the green region of Fig 6(F), the yellow region of Fig 7(F), the green region of Fig 8(F), the green region of Fig 9 (F), it can be seen that the filter also has a good ability to retain details. However, as can be seen in the yellow region of Fig 8(F), Lee-Enhanced is less capable of handling homogeneous regions.

Lee-Sigma: It can adaptively adjust the filter kernel size based on the standard deviation. However, when processing more complex images, the significant grayscale differences between pixels within the kernel often result in a generally smaller filter kernel, leading to incomplete noise removal and blurred edges. As shown by Figs 6–9(B), the image after Lee-Sigma processing will retain more strong scattering points, which retains more original information. However, the image after this filtering process is blurred, and the details of the image are lost more seriously, as shown in the green area of Fig 6(G), the details of this part have basically disappeared. At the same time, as shown in Figs 6(G) and 7(G), the contrast of the image after Lee-Sigma processing is reduced, which proves that the filter can not deal with the edge areas with large differences in gray scale.

MSF-BF: As shown by Fig 6(H)–6(J), the overall view of the urban image after MSF processing is the best, and the roads in the green and blue regions are shown clearly, and the width of the processed roads is almost the same as the original image with almost no blurring phenomenon, which proves that this algorithm has a better ability to maintain the edges. In the blue region of the water body part of the gray scale difference is small, proving that this paper's algorithm has better processing ability for homogeneous region. From the yellow part of Fig 7(H)–7(J), it can be seen that the road surface is clearly distinguished from the water body, which proves that this paper's algorithm also has good processing ability for the region with obvious and frequent gray scale changes. From the green part of Fig 8(H)–8(J), it can be seen that after denoising, the boundaries between the image farmland are clear and standardized, which is very consistent with the feature characteristics, and some of the strong feature points are also retained to a certain extent, and the yellow part of the homogeneous region, which can be seen in the figure, the overall filtering effect is better, and once again proves that

the algorithm has a better homogeneous region processing ability. From the green area of Fig 9(H)–9(J), it can be seen that the ridges after MSF-BF processing are continuous and clear, the view is clearer, there is no obvious blurring phenomenon, and the details are well preserved, and in the blue part of the valley, the distinction between the village and the river is obvious.

In conclusion, bilateral effectively suppresses noise and retains some details, but the edges of the processed image do not align with the real features, and complex regions post-filtering exhibit a smoothed shape. NLM handles simpler images well but struggles with more complex images containing finer textures. Kuan is less effective in homogeneous regions. The Lee performs weakly on edge contours, while the Lee-Enhanced excels in preserving edge contours but is less effective in homogeneous areas. The Lee-Sigma retains more original information, but the overall image appears blurry, with poor detail preservation. The 3D-NLE-BF performs well in homogeneous regions, offering improved edge contours, though it struggles with complex images that contain finer details and textures. Overall, the 3D-NLE-BF shows superior performance in homogeneous areas, with sharper edge contours that align closely with real features, while also preserving strong scattering regions to some extent.

**4.1.3. Comparison of filtering result indicators.** Due to the lack of noiseless SAR images, we used ENL [38] (Equivalent Number of Looks) and SSI [39] (Speckle Suppression Index) without reference metrics to evaluate the quality of the images.

*(1) Evaluation of denoising results in homogeneous regions.* ENL is widely used to measure the noise level of homogeneous regions; the larger the ENL, the lower the noise level of the image. In this experiment, we selected several homogeneous regions for evaluation, which have been labeled in the figures above. The ENL of these homogeneous regions is calculated using the following equation:

$$ENL = S_C\left(\frac{\mu^2}{\sigma^2}\right) \tag{32}$$

where $\mu$ and $\sigma$ denote the mean and variance of the pixels in the area to be evaluated, respectively, and the value of $S_C$ depends on the type of image, following this rule:

$$S_C = \begin{cases} 1, & \text{intensity image} \\ \frac{\pi}{4} - 1, & \text{amplitude image} \end{cases} \tag{33}$$

The data utilized in this study comprises intensity images, so take $S_C = 1$.

The following table shows the ENL of the homogeneous region filtering results for each algorithm:

As can be seen from the Table 2, NLM achieves the optimal mean NLM for homogeneous regions due to the extraction of information from the global image, and improves 47.76% compared to this paper's algorithm, but a careful analysis of the ENL of each region shows that NLM is weak for homogeneous regions in complex images, and the ENL of some regions is nearly not improved. However, the proposed algorithm in this paper achieves a certain degree of improvement in all homogeneous regions, which is 204.50%, 84.88%, 85.97%, 150.64%, and 85.43% compared to Bilateral, Kuan, Lee, Lee-Enhanced, and Lee-Sigma, respectively, proving that the algorithm in this paper has a stable and reliable homogeneous region processing capability.

*(2) Evaluation of complete image denoising results.* The metric SSI is frequently utilized to gauge the noise level in a SAR image. A lower value of SSI indicates a reduced noise level in the

**Table 2. ENL of homogeneous regions within the filtering results of each algorithm.**

| Image | Region | Noisy | Bilateral | NLM | Kuan | Lee | Lee Enhanced | Lee Sigma | 3D-NLE- BF -2 | 3D-NLE- BF -4 | 3D-NLE- BF -6 |
|---|---|---|---|---|---|---|---|---|---|---|---|
| urban | 1 | 16.894 | 40.075 | 16.901 | 53.965 | 53.547 | 58.129 | 54.752 | 63.971 | 71.437 | **74.049** |
| | 2 | 11.561 | 27.920 | 11.592 | 31.399 | 31.072 | 34.758 | 42.228 | 43.408 | 43.524 | **47.136** |
| coastal | 1 | 17.341 | 98.267 | 66.036 | 104.849 | 104.386 | 111.921 | 144.610 | 117.465 | 159.804 | **207.306** |
| | 2 | 21.674 | 206.608 | **1310.373** | 245.402 | 243.928 | 254.132 | 403.638 | 311.839 | 501.499 | 709.696 |
| farmland | 1 | 15.331 | 44.019 | 66.036 | 104.849 | 104.386 | 49.231 | 54.621 | 117.465 | 159.804 | **207.306** |
| | 2 | 16.817 | 60.251 | **1310.373** | 245.402 | 243.928 | 71.511 | 83.709 | 311.839 | 501.499 | 709.696 |
| mountain | 1 | 12.195 | 49.033 | 13.177 | 45.020 | 43.829 | 59.087 | 79.021 | 63.525 | 76.781 | **77.333** |
| | 2 | 10.439 | 36.938 | 10.775 | 31.914 | 30.900 | 44.724 | 59.731 | 46.716 | 54.684 | **55.075** |

image. The metric SSI can be computed using the following equation:

$$SSI = \frac{u_n \sigma}{\sigma_n u} \tag{34}$$

In the above equation:

$u_n$ and $\sigma_n$ represent the mean and standard deviation of the original image pixels, respectively, while $u$ and $\sigma$ signify the mean and standard deviation of the denoised SAR image pixels.

All denoised images are assessed using SSI, with the evaluation results displayed in the table below:

As can be seen from Table 3, the average SSI of this paper's algorithm achieves the optimal result among all the filtering results, which is reduced by 16.34%, 37.45%, 15.33%, 15.93%, 9.22%, and 10.31%, respectively, in comparison with Bilateral, NLM, Kuan, Lee, Lee, Lee-Enhanced, and Lee-Sigma, proving that this paper's algorithm is also good at processing complex images., proving that the algorithms in this paper also have good processing ability for complex images.

## 4.2. Simulated SAR denoising experiment

**4.2.1. Experiment data.** From Eq 1, it can be seen that the intensity of noise is regulated by the variance v. The larger v is, the stronger the noise interferes with the image, and the worse the visual effect of the image is. Accordingly, we add different degrees of noise to the noise-free image to evaluate the performance of 3D-NLE-BF. Goldhill, Lena, and Peppers were chosen as the noise-free images for the simulation experiment, and the size of each image was 512*512 pixels. The data used in this experiment were obtained for free from Copernicus Open Access Hub (https://scihub.copernicus.eu/) In this experiment, the noise with variance 0.06, 0.08, and 0.1 was added to the images sequentially, and the original image and the noise-added image are shown in Fig 10.

**4.2.2. Results and analysis.** In Figs 11–19(A), the bilateral filtering results in significant grayscale variations within homogeneous regions, which become increasingly pronounced as

**Table 3. SSI of the filtering results for the complete image of each algorithm.**

| Image | Bilateral | NLM | Kuan | Lee | Lee Enhanced | Lee Sigma | 3D-NLE- BF -2 | 3D-NLE- BF -4 | 3D-NLE- BF -6 |
|---|---|---|---|---|---|---|---|---|---|
| urban | 0.793 | 0.999 | 0.739 | 0.741 | 0.716 | 0.738 | 0.670 | 0.656 | **0.652** |
| coastal | 0.745 | 0.855 | 0.777 | 0.784 | 0.715 | 0.745 | 0.671 | 0.657 | **0.618** |
| farmland | 0.661 | 0.994 | 0.635 | 0.638 | 0.614 | 0.609 | 0.564 | 0.519 | **0.486** |
| mountain | 0.672 | 0.992 | 0.686 | 0.694 | 0.601 | 0.586 | 0.564 | 0.558 | **0.591** |

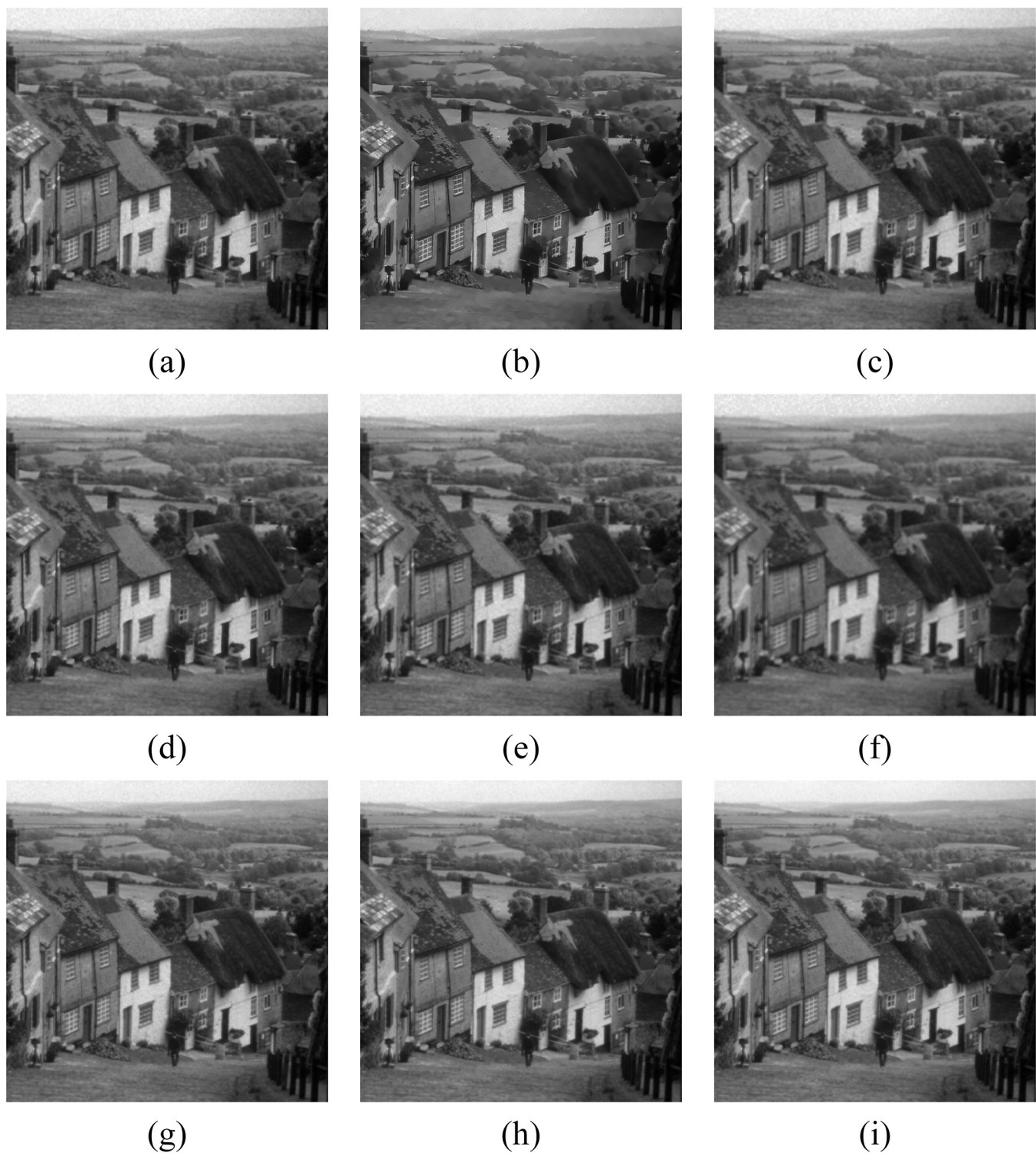

**Fig 11. Goldhill denoising results with noise variance of 0.06.** (a) Bilateral. (b) NLM. (c) Kuan. (d) Lee. (e) Lee-Enhanced. (f) Lee-Sigma. (g-i) 3D-NLE-BF with inputs of 2, 4, and 6 auxiliary images.

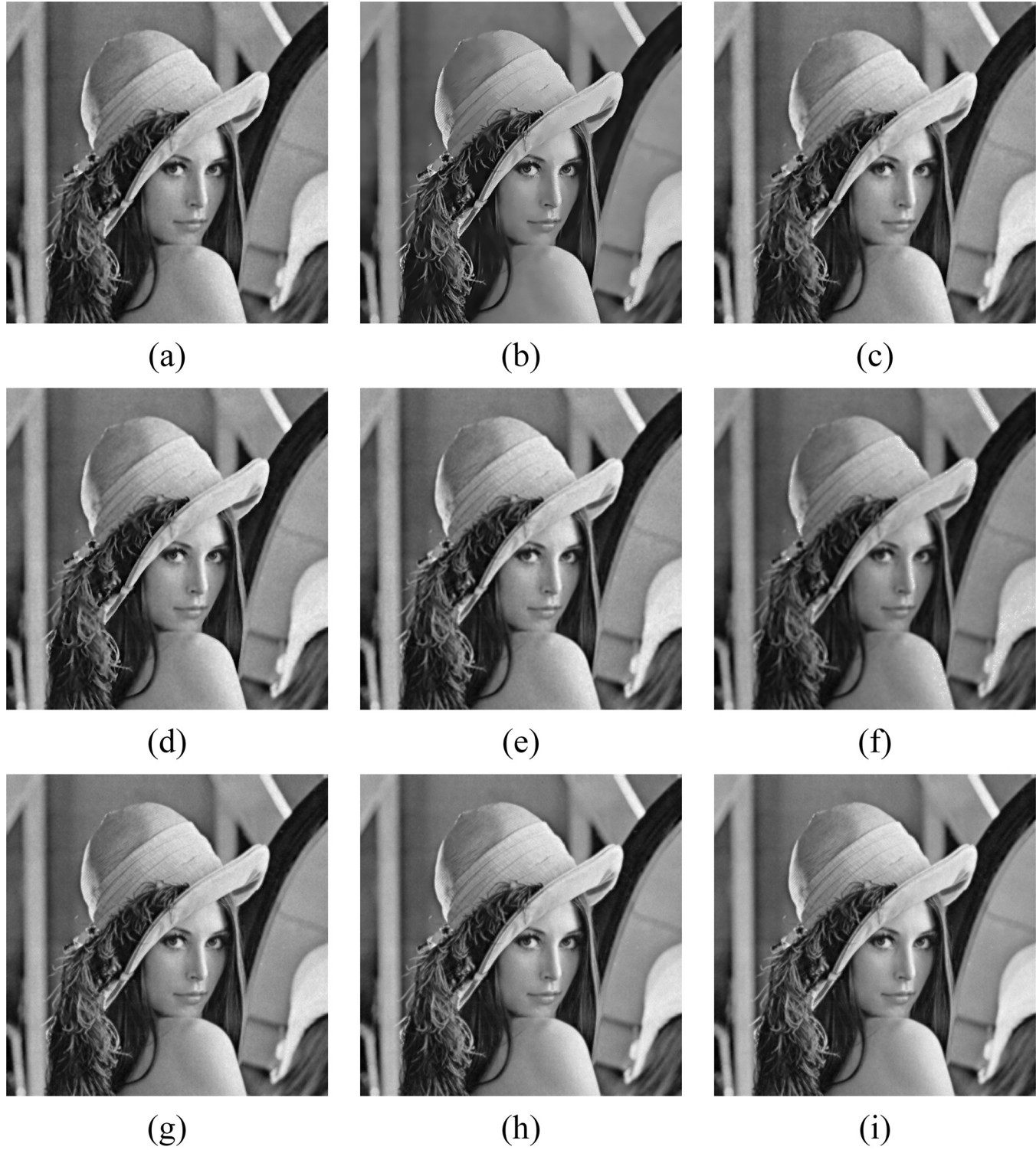

**Fig 12. Lena denoising results with noise variance of 0.06.** (a) Bilateral. (b) NLM. (c) Kuan. (d) Lee. (e) Lee-Enhanced. (f) Lee-Sigma. (g-i) 3D-NLE-BF with inputs of 2, 4, and 6 auxiliary images.

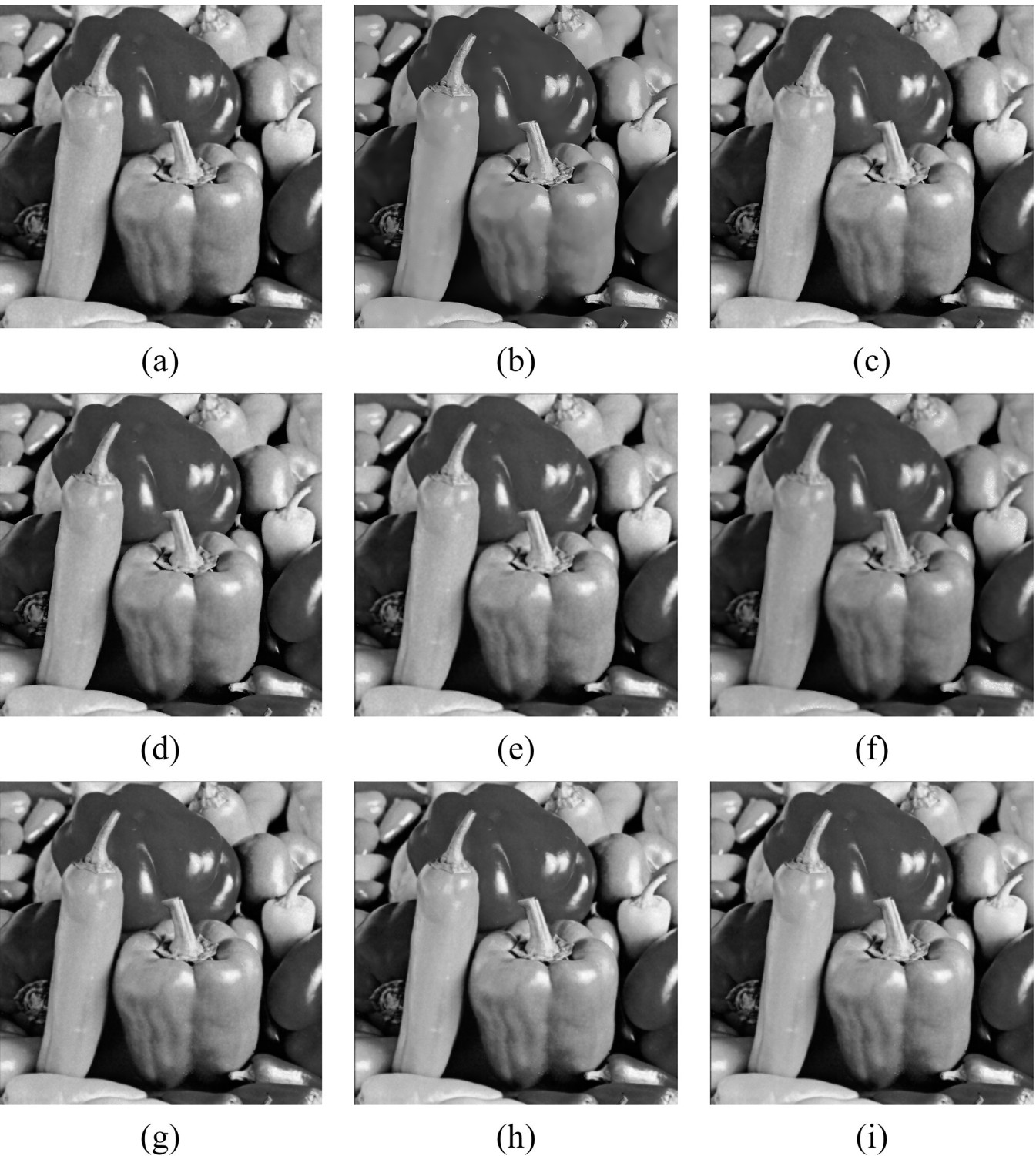

**Fig 13. Peppers denoising results with noise variance of 0.06.** (a) Bilateral. (b) NLM. (c) Kuan. (d) Lee. (e) Lee-Enhanced. (f) Lee-Sigma. (g-i) 3D-NLE-BF with inputs of 2, 4, and 6 auxiliary images.

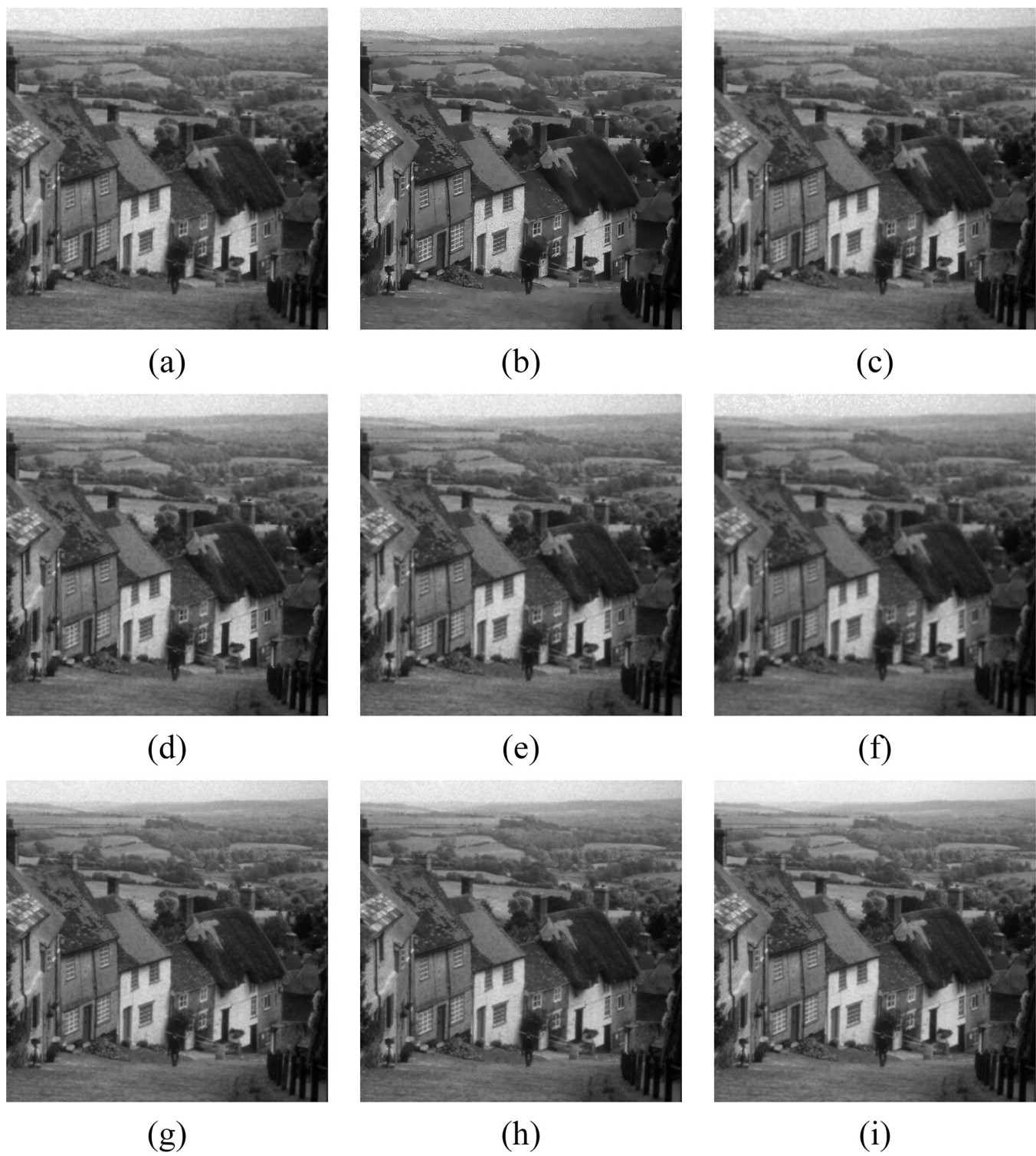

**Fig 14. Goldhill denoising results with noise variance of 0.08.** (a) Bilateral. (b) NLM. (c) Kuan. (d) Lee. (e) Lee-Enhanced. (f) Lee-Sigma. (g-i) 3D-NLE-BF with inputs of 2, 4, and 6 auxiliary images.

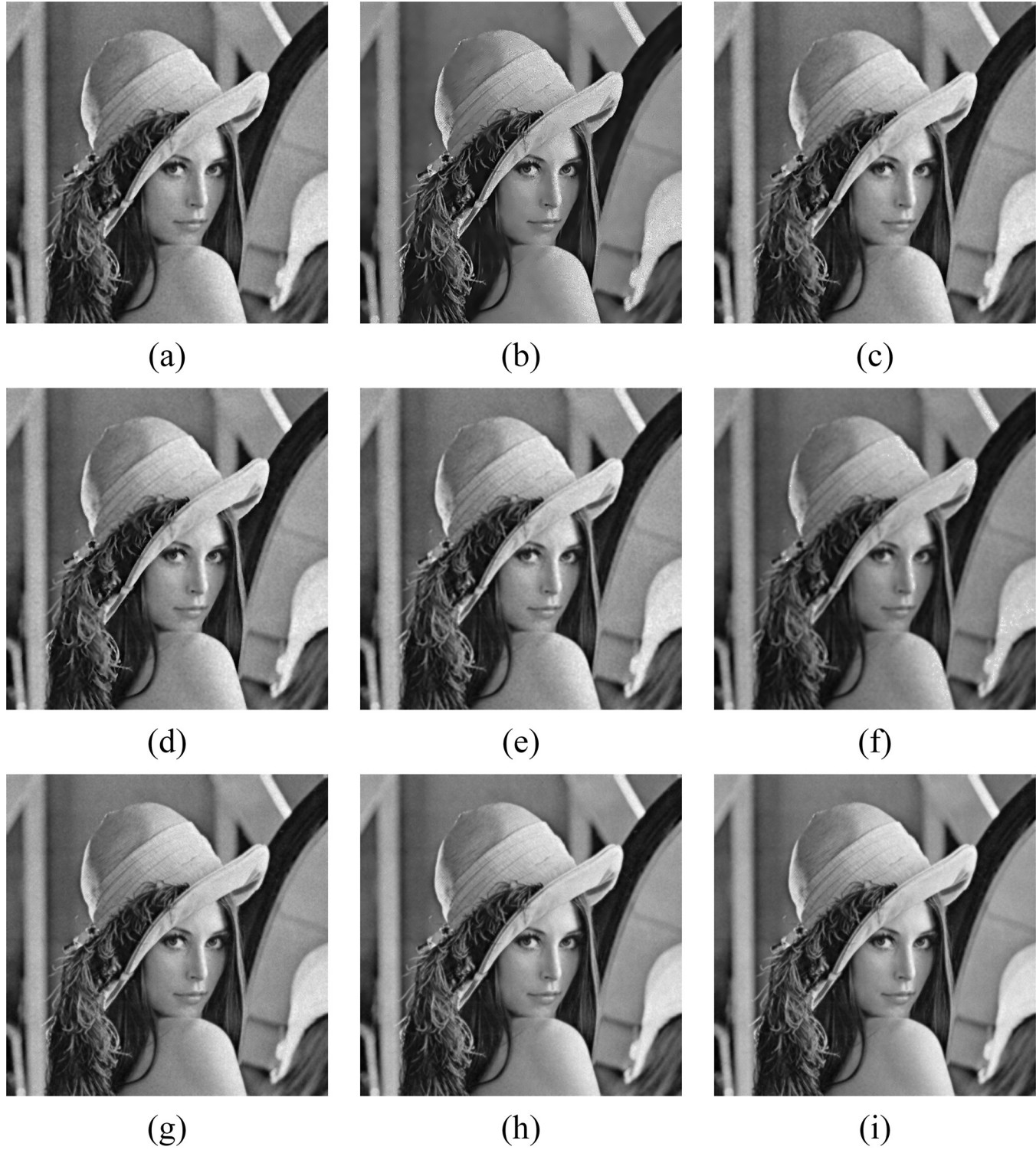

**Fig 15. Lena denoising results with noise variance of 0.08.** (a) Bilateral. (b) NLM. (c) Kuan. (d) Lee. (e) Lee-Enhanced. (f) Lee-Sigma. (g-i) 3D-NLE-BF with inputs of 2, 4, and 6 auxiliary images.

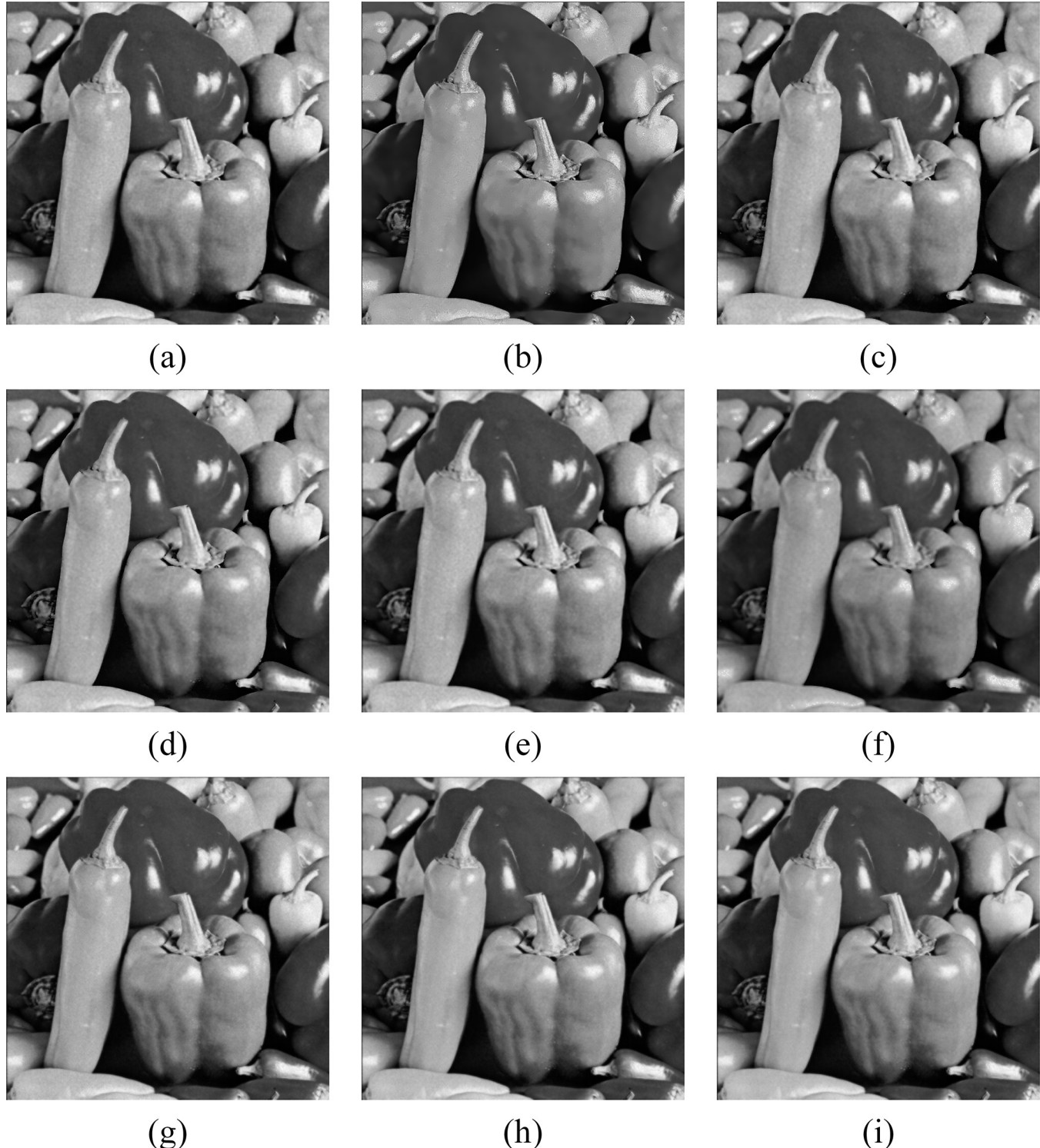

**Fig 16. Peppers denoising results with noise variance of 0.08.** (a) Bilateral. (b) NLM. (c) Kuan. (d) Lee. (e) Lee-Enhanced. (f) Lee-Sigma. (g-i) 3D-NLE-BF with inputs of 2, 4, and 6 auxiliary images.

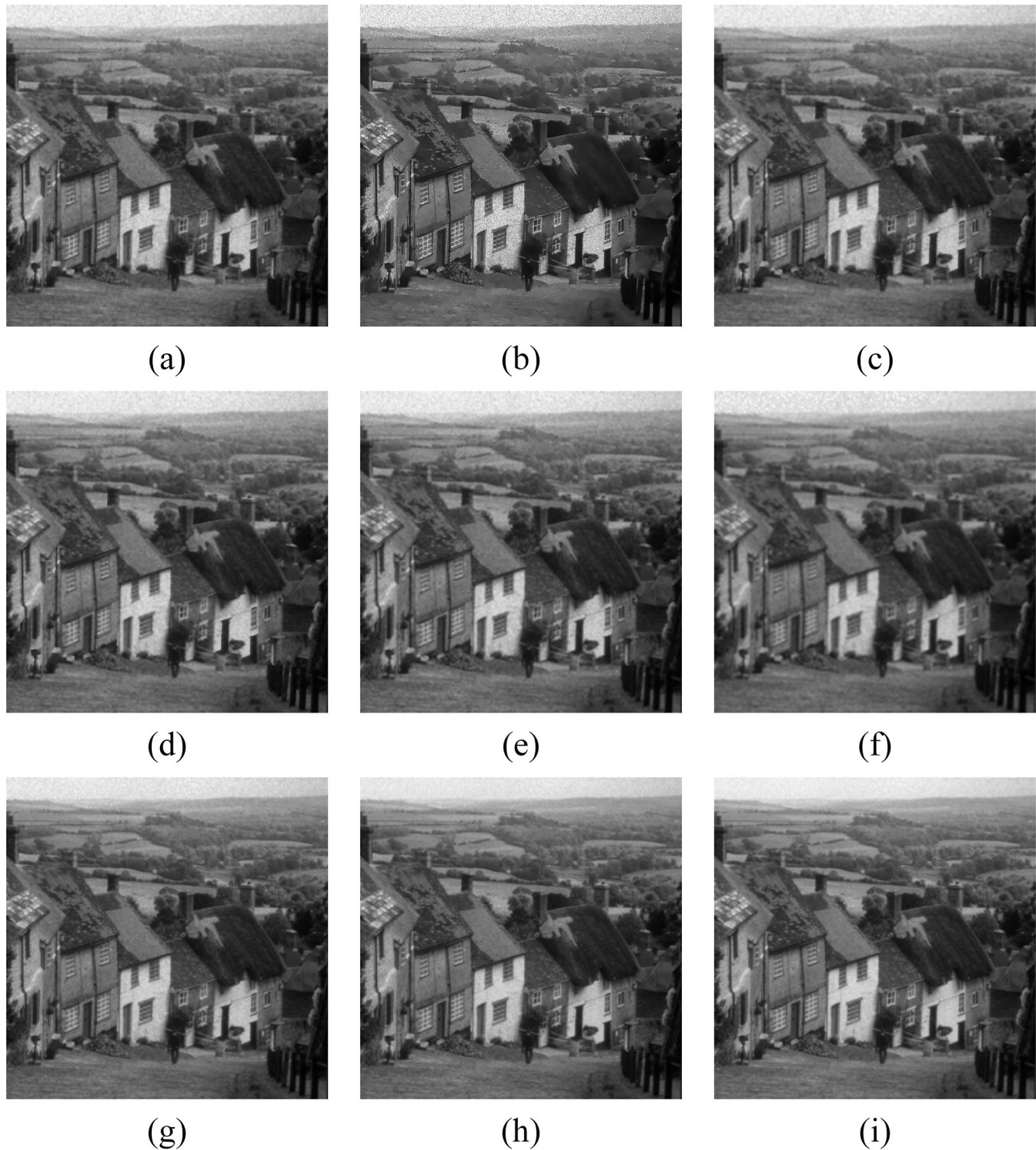

**Fig 17. Goldhill denoising results with noise variance of 0.1.** (a) Bilateral. (b) NLM. (c) Kuan. (d) Lee. (e) Lee-Enhanced. (f) Lee-Sigma. (g-i) 3D-NLE-BF with inputs of 2, 4, and 6 auxiliary images.

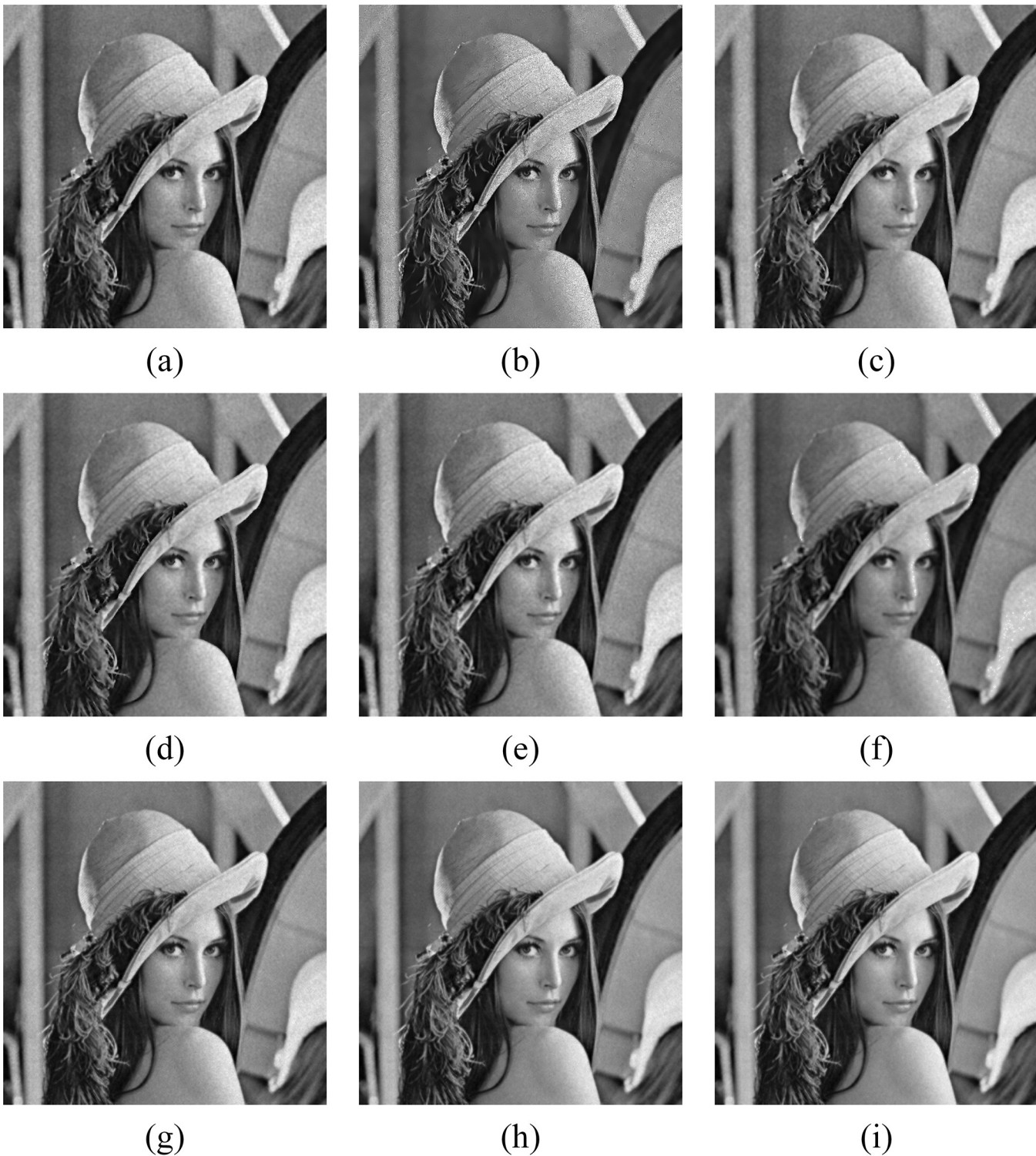

**Fig 18. Lena denoising results with noise variance of 0.1.** (a) Bilateral. (b) NLM. (c) Kuan. (d) Lee. (e) Lee-Enhanced. (f) Lee-Sigma. (g-i) 3D-NLE-BF with inputs of 2, 4, and 6 auxiliary images.

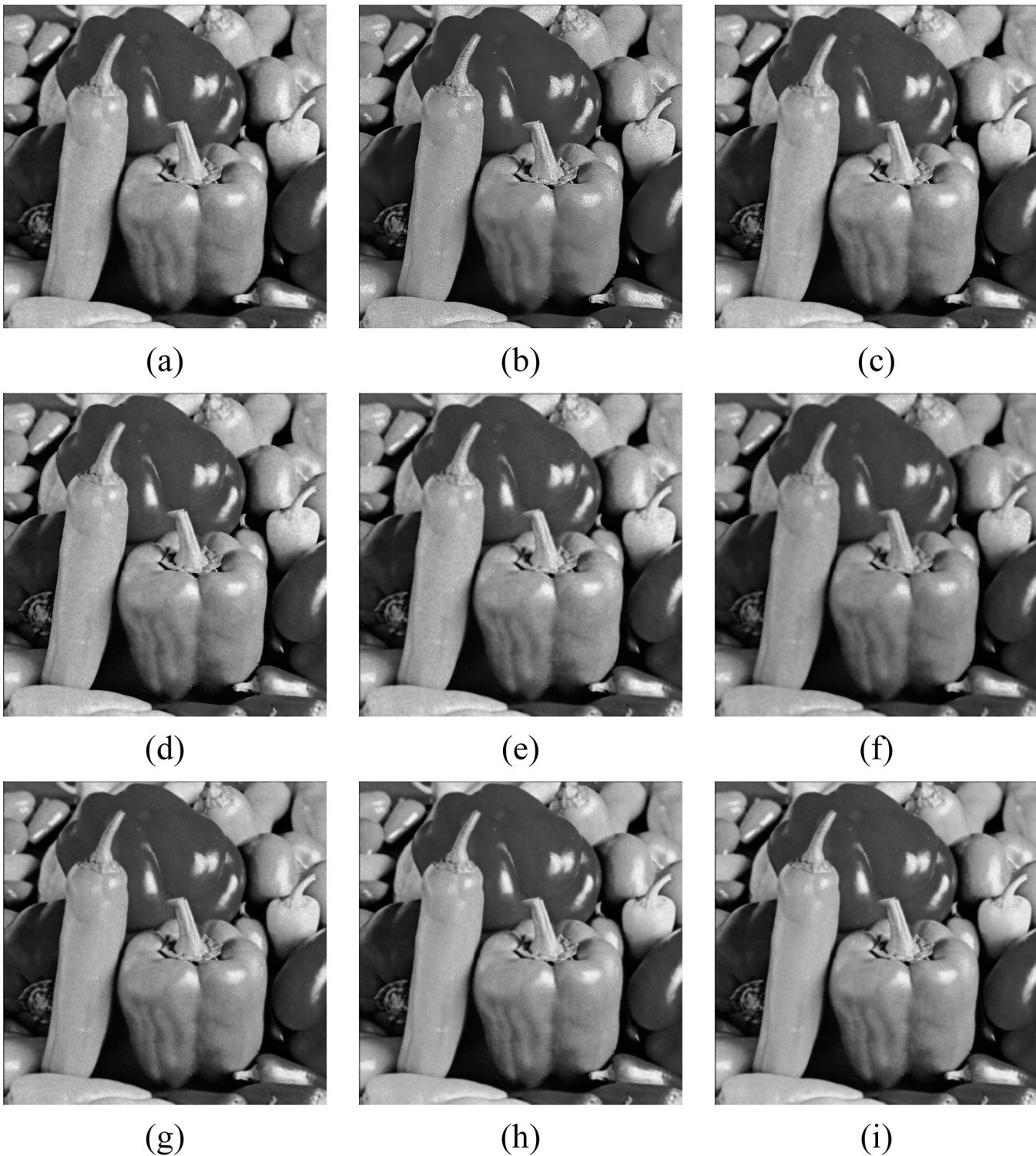

**Fig 19. Peppers denoising results with noise variance of 0.1.** (a) Bilateral. (b) NLM. (c) Kuan. (d) Lee. (e) Lee-Enhanced. (f) Lee-Sigma. (g-i) 3D-NLE-BF with inputs of 2, 4, and 6 auxiliary images.

noise intensity rises, along with a degree of edge blurring. In Figs 11–19(B), for images with low noise intensity, NLM maintains stable grayscale within homogeneous regions and provides clear edge contours. However, as noise intensity increases, NLM's denoising ability diminishes, leading to notable information loss in the processed image.

Figs 11–19(C) shows that Kuan filtering has limited effectiveness in homogeneous regions, with this limitation becoming more evident as noise intensity grows. In Figs 11–19(D), Lee filtering is heavily influenced by the local statistical values of pixels within the kernel. Consequently, high-noise images processed with Lee suffer substantial detail loss, though performance improves in lower-noise conditions. Figs 11–19(E) shows that Lee-Enhanced filtering results in increased blurring and more severe loss of original information. Figs 11–19(F) indicates that the Lee-Sigma filter better preserves high-brightness pixels, reflecting improved retention of original information. However, with higher noise levels, the filtered image becomes progressively blurred.

Finally, Figs 11–19(G)–19(I) demonstrate that the proposed algorithm maintains a consistent grayscale within homogeneous regions and effectively preserves edge information across varying noise intensities. The denoised image remains clearer and more stable under different noise intensities, indicating the robustness of the filtering effect.

**4.2.3. Comparison of filtering result indicators.** In this section, we use two common metrics to measure the denoising results of the simulated SAR images, PSNR and UIQI [40].

Peak signal-to-noise ratio (PSNR) is often used as a measure of the difference between two images, the larger the PSNR, the higher the degree of similarity between the two images, the PSNR is calculated as shown below:

$$PSNR = 10 \cdot \log_{10} \left( \frac{MAX^2}{MSE} \right) \tag{35}$$

In the above equation, $MAX$ is the gray scale range, which is taken as 255 in this paper, and $MSE$ is the mean square error of the two images, which is calculated as shown below:

$$MSE = \frac{1}{mn} \cdot \sum_{i=1}^{m} \sum_{j=1}^{n} \left( x_{i,j} - y_{i,j} \right)^2 \tag{36}$$

In the above equation, $m,n$ denotes the number of rows and columns of the image, $x,y$ denotes the noise-free image and the denoised image, respectively, and $i,j$ denotes the position coordinates.

The PSNR of the denoised images processed by each algorithm is shown in the following table:

The UIQI takes into account the correlation between the image to be evaluated and the reference image, the difference in gray scale and the difference in contrast, and can be used to evaluate the quality of the denoised image, the larger the UIQI is, the higher the degree of similarity between the denoised image and the original image is, and the better the quality of the denoised image is, and the calculation of the UIQI is shown as follows:

$$UIQI = \frac{\sigma_{xy}}{\sigma_x \sigma_y} \cdot \frac{2\mu_x \mu_y}{\mu_x^2 + \mu_y^2} \cdot \frac{2\sigma_x \sigma_y}{\sigma_x^2 + \sigma_y^2} \tag{37}$$

In the above equation, $x,y$ denotes the denoised image and the noiseless image, $\sigma_x, \sigma_y, \sigma_{xy}$ denotes the variance of $x$, the variance of $y$, the covariance of $x,y$, respectively, and $\mu_x, \mu_y$ denotes the mean value of $x,y$.

The UIQI of the denoised images processed by each algorithm is shown in the following table:

**Table 4. PSNR of the results of each algorithm.**

| noise | Image | Bilateral | NLM | Kuan | Lee | Lee Enhanced | Lee Sigma | 3D-NLE- BF -2 | 3D-NLE- BF -4 | 3D-NLE- BF -6 |
|-------|-------|-----------|-----|------|-----|--------------|-----------|---------------|---------------|---------------|
| 0.06 | Goldhill | 30.703 | **33.162** | 29.128 | 29.160 | 28.428 | 27.997 | 32.071 | 32.253 | 32.331 |
| | Lena | 32.600 | **35.262** | 31.337 | 31.414 | 29.533 | 29.461 | 32.794 | 33.026 | 33.129 |
| | Peppers | 32.677 | **34.798** | 32.199 | 32.318 | 29.365 | 28.832 | 33.022 | 33.218 | 33.308 |
| 0.08 | Goldhill | 30.496 | 30.987 | 29.026 | 29.058 | 28.301 | 27.913 | 31.762 | 32.038 | **32.148** |
| | Lena | 32.160 | 31.970 | 31.099 | 31.171 | 29.356 | 29.313 | 32.597 | 32.997 | **33.160** |
| | Peppers | 32.301 | 32.273 | 31.900 | 32.006 | 29.218 | 28.711 | 32.702 | 33.020 | **33.149** |
| 0.1 | Goldhill | 30.211 | 28.299 | 28.920 | 28.954 | 28.157 | 27.818 | 31.449 | 31.862 | **32.036** |
| | Lena | 31.592 | 28.206 | 30.777 | 30.842 | 29.103 | 29.141 | 32.087 | 32.623 | **32.859** |
| | Peppers | 31.864 | 29.010 | 31.608 | 31.699 | 29.074 | 28.587 | 32.321 | 32.778 | **32.976** |

As can be seen from Tables 4 and 5, when the noise content is low, the NLM processed image achieves the optimal denoising result, but as the noise content rises, the filtering effect of NLM becomes worse and worse. Whereas, the filtering effect of 3D-NLE-BF has remained stable and has achieved optimal filtering results in high noise images. Compared to Bilateral, NLM, Kuan, Lee, Lee-Enhanced, and Lee-Sigma, the PSNR of the algorithm proposed in this paper is improved by: 3.03%, 3.27%, 6.25%, 6.01%, 12.55%, and 13.76%, and the UIQI is improved by 0.19%, 0.31%, 0.46%, 0.45%, 0.95% and 1.08% respectively.

## 4.3. Runtime comparison

Assume that the input image size is $N \times N$ with a total of $M$ images. Since the value of $M$ is usually small and can be treated as a constant, the time complexity of the algorithm in this paper is $O(N^2)$.

All the experimental results are assessed in Python version = 3.11 on AMD Ryzen 7 5800H CPU @ 3.2 GHz, 8 GB RAM and 64-bit operating system. The Fig 20 shows the average running time of each algorithm:

As can be seen from Fig 20, the computation time of 3D-NLE-BF is long and the running time shows a linear positive correlation with the number of images involved in the computation. To reduce the time cost of this algorithm, it is recommended that the number of auxiliary images be set to 4.

## 5. Conclusion and prospects

In this paper, we propose the XX algorithm, which uses the multi-temporal SAR centered on the target temporal phase as input, evaluates noise content based on the temporal and spatial

**Table 5. UIQI of the results of each algorithm.**

| noise | Image | Bilateral | NLM | Kuan | Lee | Lee Enhanced | Lee Sigma | 3D-NLE- BF -2 | 3D-NLE- BF -4 | 3D-NLE- BF -6 |
|-------|-------|-----------|-----|------|-----|--------------|-----------|---------------|---------------|---------------|
| 0.06 | Goldhill | 0.988 | **0.994** | 0.983 | 0.983 | 0.980 | 0.978 | 0.992 | 0.992 | 0.992 |
| | Lena | 0.992 | **0.996** | 0.989 | 0.990 | 0.984 | 0.983 | 0.992 | 0.993 | 0.993 |
| | Peppers | 0.995 | **0.997** | 0.994 | 0.994 | 0.988 | 0.987 | 0.995 | 0.995 | 0.995 |
| 0.08 | Goldhill | 0.988 | 0.989 | 0.983 | 0.983 | 0.980 | 0.978 | 0.991 | 0.991 | **0.992** |
| | Lena | 0.991 | 0.991 | 0.989 | 0.989 | 0.983 | 0.983 | 0.992 | 0.993 | **0.993** |
| | Peppers | 0.994 | 0.994 | 0.994 | 0.994 | 0.988 | 0.986 | 0.995 | 0.995 | **0.995** |
| 0.1 | Goldhill | 0.987 | 0.980 | 0.982 | 0.982 | 0.979 | 0.977 | 0.990 | 0.991 | **0.991** |
| | Lena | 0.990 | 0.979 | 0.988 | 0.988 | 0.982 | 0.982 | 0.991 | 0.992 | **0.992** |
| | Peppers | 0.994 | 0.988 | 0.993 | 0.993 | 0.988 | 0.986 | 0.994 | 0.995 | **0.995** |

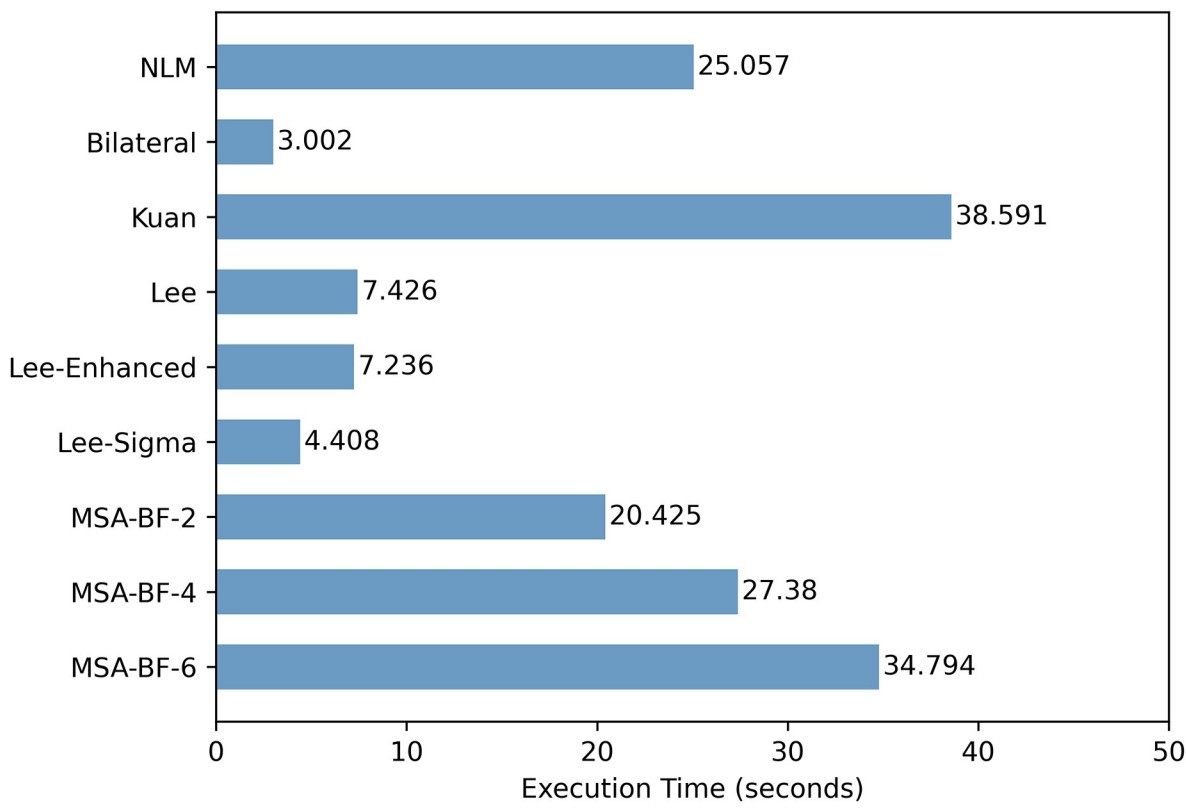

**Fig 20. Running time of each algorithm.**

stability of each pixel, and categorizes pixels as strong-noise, weak-noise, or noise-free accordingly. Tailored filtering methods are applied for each pixel type, effectively addressing the issue of information loss caused by a single filtering method. To further reduce irrelevant information in denoised pixels, the algorithm incorporates range-weight, spatial-weight, confidence-weight, and time-weight, and designs filter kernels for strong- and weak-noise pixels based on these four weights. This approach leverages the abundant information in multi-temporal SAR, effectively reducing image blurring and enhancing edge clarity. The range-weight uses the SAR pixel similarity criterion, enabling the bilateral filter to effectively manage SAR's multiplicative noise, further improving SAR image quality.

To verify the effectiveness of the proposed algorithm, we compare it with Bilateral, NLM, Kuan, Lee, Lee-Enhanced, and Lee-Sigma filters in both real and simulated SAR denoising experiments. In real SAR experiments, we use ENL and SSI metrics to evaluate each algorithm's performance in homogeneous and complex regions. Results indicate that the ENL of images processed by our algorithm is improved by 204.50%, 84.88%, 85.97%, 150.64%, 85.43%, and the SSI reduced by 16.34%, 37.45%, 15.33%, 15.93%, 9.22%, 10.31%, compared to other algorithms, demonstrating superior denoising capability in both region types. In simulated SAR experiments, we apply varying levels of noise to noiseless images, showing that the proposed algorithm's filtering effect remains stable despite noise intensity, delivering reliable results in both high- and low-noise conditions. Additionally, PSNR and UIQI metrics are used to evaluate each algorithm's denoised images, with our algorithm achieving an average PSNR improvement of 3.03%, 3.27%, 6.25%, 6.01%, 12.55%, 13.76% and UIQI improvement of 0.19%, 0.31%, 0.46%, 0.45%, 0.95%, 1.08% over other methods.

In conclusion, the algorithm presented in this paper demonstrates excellent filtering capabilities. It effectively removes noise from conventional SAR images, significantly enhancing image quality and retaining more original details. Nonetheless, it has certain limitations: its reliance on multiple SAR images for auxiliary denoising increases computation time compared to other methods. Additionally, the strict criteria for identifying noise-free pixels result in some loss of original information in the denoised output. Future work could focus on refining the identification of noise-free pixels to enhance image quality and on optimizing the denoising process to reduce computational time.

## Author Contributions

**Conceptualization:** Haiyan Zhang, Guoyin Cai.

**Data curation:** Haiyan Zhang, Yang Liu.

**Formal analysis:** Haiyan Zhang.

**Funding acquisition:** Guoyin Cai.

**Investigation:** Guoyin Cai.

**Methodology:** Haiyan Zhang.

**Resources:** Haiyan Zhang.

**Software:** Haiyan Zhang.

**Supervision:** Yang Liu, Guoyin Cai.

**Validation:** Haiyan Zhang, Yang Liu, Guoyin Cai.

**Writing – original draft:** Haiyan Zhang.

**Writing – review & editing:** Yang Liu, Guoyin Cai.

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
