## [Decision Letter · Decision Letter 0]

12 Oct 2024

PONE-D-24-36350A Novel Denoising Method Based on Bilateral Filter with Multitemporal SAR-AssistedPLOS ONE

Dear Dr. Cai,

Thank you for submitting your manuscript to PLOS ONE. After careful consideration, we feel that it has merit but does not fully meet PLOS ONE’s publication criteria as it currently stands. Therefore, we invite you to submit a revised version of the manuscript that addresses the points raised during the review process.

We look forward to receiving your revised manuscript.

Kind regards,

Khan Bahadar Khan, Ph.D

Academic Editor

PLOS ONE

Journal Requirements:

3. We note that Figures 1, 3, 4, 5, 7, 9, and 10 in your submission contain [map/satellite] images which may be copyrighted. All PLOS content is published under the Creative Commons Attribution License (CC BY 4.0), which means that the manuscript, images, and Supporting Information files will be freely available online, and any third party is permitted to access, download, copy, distribute, and use these materials in any way, even commercially, with proper attribution. For these reasons, we cannot publish previously copyrighted maps or satellite images created using proprietary data, such as Google software (Google Maps, Street View, and Earth). For more information, see our copyright guidelines: http://journals.plos.org/plosone/s/licenses-and-copyright.

a. You may seek permission from the original copyright holder of Figures 1, 3, 4, 5, 7, 9, and 10 to publish the content specifically under the CC BY 4.0 license.  

4. Please ensure that you refer to Figure 2, 3, and 4 in your text as, if accepted, production will need this reference to link the reader to the figure.

Reviewers' comments:

Reviewer's Responses to Questions

**Comments to the Author**

1. Is the manuscript technically sound, and do the data support the conclusions?

Reviewer #1: Partly

Reviewer #2: No

Reviewer #3: Partly

Reviewer #4: Yes

2. Has the statistical analysis been performed appropriately and rigorously? 

Reviewer #1: Yes

Reviewer #2: Yes

Reviewer #3: No

Reviewer #4: Yes

3. Have the authors made all data underlying the findings in their manuscript fully available?

Reviewer #1: Yes

Reviewer #2: Yes

Reviewer #3: No

Reviewer #4: Yes

4. Is the manuscript presented in an intelligible fashion and written in standard English?

Reviewer #1: Yes

Reviewer #2: Yes

Reviewer #3: No

Reviewer #4: Yes

5. Review Comments to the Author

Reviewer #1: Explain how the present manuscript differs from and improves upon these established previous works. Include a study with your survey papers and be sure to highlight any knowledge gaps that have come up recently.

Check and if in case missing, discuss some major sections in the article like the real time/ practical application of the paper, future perspectives, major contribution in the paper, significance of this study.

In the introduction section, include some other research applications as literature. Some suggestions are: https://ieeexplore.ieee.org/abstract/document/10411937 ; https://link.springer.com/article/10.1007/s11831-021-09548-z ; https://ieeexplore.ieee.org/abstract/document/7748978 ; https://www.sciencedirect.com/science/article/pii/S2215098617314003 ; https://ieeexplore.ieee.org/abstract/document/10447792

Reviewer #2: work done is very good but the time complexity also must be taken into account ,otherwise it time for preprocessing is more than actual method the proposed method cannot be feasible in applications of GEOspatial techniques

Reviewer #3: The manuscript entitled “A Novel Denoising Method Based on Bilateral Filter with Multitemporal SAR-Assisted” has been investigated in detail. This manuscript introduces the Multitemporal SAR-Assisted Bilateral Filtering (MSA-BF) algorithm, which aims to improve denoising in SAR images by leveraging multi-temporal data. While the proposed approach shows promise, the paper suffers from a lack of clear justification, overly simplified pixel classification, and limited theoretical depth. The experimental methodology and results are insufficiently detailed, and the claims made regarding the performance of the algorithm appear exaggerated without rigorous statistical backing. Moreover, the omission of computational complexity analysis and the poor structure of the results section further weaken the manuscript. A major revision is necessary to address these critical issues before the manuscript can be considered for publication.

1) The manuscript introduces the Multitemporal SAR-Assisted Bilateral Filtering (MSA-BF) algorithm but fails to provide a strong justification for why this approach is superior to other well-established filters for SAR images. While the paper highlights some of the limitations of the bilateral filter, such as its inability to handle multiplicative noise, the discussion lacks depth regarding the limitations of other comparison filters (e.g., NLM, Lee). A more thorough review of the existing methods and clearer arguments for the necessity of the proposed algorithm are needed.

2) The proposed algorithm introduces several new components, including confidence-weight, time-weight, and pixel similarity evaluation for range-weight computation. However, the theoretical background and rationale behind these components are not sufficiently explained. The manuscript should include detailed descriptions of how these weights are calculated, their mathematical formulations, and the reasoning behind the choice of these specific techniques.

3) The manuscript classifies pixels into three categories—strong noise, weak noise, and noise-free pixels—based on noise content along the time axis. However, this approach seems overly simplistic, particularly for handling complex SAR images where noise and signal characteristics may vary significantly across both time and spatial dimensions. A more sophisticated and adaptive classification scheme may be necessary to handle the diverse noise characteristics in SAR data more effectively.

4) The experimental section does not provide sufficient details on how the comparisons between the MSA-BF algorithm and other filters were conducted. For example, the manuscript does not specify the exact parameters used for each comparison filter or the specifics of the multi-temporal SAR data sets used. Clear descriptions of the experimental setup, including the choice of dataset and parameter tuning for each filter, are essential for ensuring the reproducibility of the results.

5) While the manuscript reports improvements in ENL (Equivalent Number of Looks) and SSI (Structural Similarity Index), there is little discussion of why these metrics were chosen and what they reveal about the algorithm’s performance. Moreover, the results would benefit from additional quantitative metrics such as PSNR (Peak Signal-to-Noise Ratio) and visual comparisons to provide a more comprehensive evaluation of the filtering performance.

6) “Discussion” section should be added in a more highlighting, argumentative way. The author should analysis the reason why the tested results is achieved.

7) The authors should clearly emphasize the contribution of the study. Please note that the up-to-date of references will contribute to the up-to-date of your manuscript. The studies named- “Artificial intelligence-based robust hybrid algorithm design and implementation for real-time detection of plant diseases in agricultural environments; Detection of solder paste defects with an optimization‐based deep learning model using image processing techniques”- can be used to explain the methodology and highlight the performance in the study or to indicate the contribution in the “Introduction” section.

8) The claim that the proposed algorithm improves filtering in homogeneous regions "by nearly threefold" and enhances performance in large complex regions "by approximately 20%" seems exaggerated without adequate support. The manuscript should provide more rigorous statistical evidence and error analysis to substantiate these claims. Otherwise, the paper risks overclaiming the benefits of the proposed approach.

9) The manuscript does not provide any analysis of the computational complexity of the MSA-BF algorithm. Given that the algorithm introduces several new components, such as confidence-weight and time-weight, it is important to discuss the computational trade-offs involved in achieving the reported performance improvements. An analysis of runtime or resource consumption compared to other filters would be a valuable addition.

10) The results section is not well-structured, making it difficult to follow the comparisons and assess the performance of the MSA-BF algorithm. The paper would benefit from organizing the results into clearer subsections with tables, graphs, and visual comparisons for each Earth surface type. This would improve readability and allow for a more transparent presentation of the findings.

Reviewer #4: The contribution of this manuscript is clear and good. Therefore, my recommendation is to accept (Minor Revision) this manuscript that has Ref. No.: PONE-D-24-36350 after several adjustments must be made before publication.

My specific comments are:

1- The author did not describe the drawbacks of each conventional technique in the introduction paragraph.

2- Add the problem definition of this work in the introduction paragraph because it is not clear.

3- Please include the references for all equations.

6. PLOS authors have the option to publish the peer review history of their article (what does this mean?). If published, this will include your full peer review and any attached files.

Reviewer #1: No

Reviewer #2: No

Reviewer #3: No

Reviewer #4: No

---

## [Author Response · Author response to Decision Letter 0]

15 Nov 2024

1. [Editor's comment]: We have revised the manuscript's format, uploaded the experimental data, and provided a clear description of the data sources.

2. [Reviewer 1's comment]: We have re-summarized the shortcomings of existing algorithms, emphasizing the design principles and advantages of the proposed algorithm. Additionally, we have cited the recommended references in the manuscript.

3. [Reviewer 2's comment]: We have added an analysis of the time complexity of the proposed algorithm in the manuscript.

4. [Reviewer 3's comment]: We have supplemented and summarized the shortcomings of existing algorithms, incorporated the recommended references, and provided a more detailed description of the four components of the proposed algorithm. We also redesigned the pixel classification rule based on the time and spatial stability of the pixels to be denoised. Furthermore, we included an analysis of the algorithm's complexity, designed a simulated SAR denoising experiment, and evaluated the results using PSNR and UIQI. Lastly, we further refined the result analysis and other sections of the manuscript.

5. [Reviewer 4's comment]: We have analyzed and summarized the current state of the technology and clarified the specific problems addressed by the proposed algorithm. Finally, we made revisions to the manuscript's format and citations.

---

## [Decision Letter · Decision Letter 1]

26 Nov 2024

A Novel 3D Bilateral Filtering Algorithm with Noise Level Estimation Assisted by Multi-Temporal SAR

PONE-D-24-36350R1

Dear Dr. Cai,

We’re pleased to inform you that your manuscript has been judged scientifically suitable for publication and will be formally accepted for publication once it meets all outstanding technical requirements.

Kind regards,

Khan Bahadar Khan, Ph.D

Academic Editor

PLOS ONE

Additional Editor Comments (optional):

Reviewers' comments:

Reviewer's Responses to Questions

**Comments to the Author**

1. If the authors have adequately addressed your comments raised in a previous round of review and you feel that this manuscript is now acceptable for publication, you may indicate that here to bypass the “Comments to the Author” section, enter your conflict of interest statement in the “Confidential to Editor” section, and submit your "Accept" recommendation.

Reviewer #1: All comments have been addressed

Reviewer #4: All comments have been addressed

2. Is the manuscript technically sound, and do the data support the conclusions?

Reviewer #1: Yes

Reviewer #4: Yes

3. Has the statistical analysis been performed appropriately and rigorously? 

Reviewer #1: Yes

Reviewer #4: Yes

4. Have the authors made all data underlying the findings in their manuscript fully available?

Reviewer #1: Yes

Reviewer #4: Yes

5. Is the manuscript presented in an intelligible fashion and written in standard English?

Reviewer #1: Yes

Reviewer #4: Yes

6. Review Comments to the Author

Reviewer #1: The authors have thoroughly addressed all the comments provided in the review. Each point has been incorporated effectively, demonstrating their attention to detail and commitment to improving the manuscript.

The revisions align well with the suggestions, and the responses provided by the authors are clear and satisfactory. The updated version reflects significant improvements in terms of clarity, content, and overall quality. Additionally, the authors have ensured that all necessary changes have been implemented without introducing new issues. The manuscript in its current form meets the required standards for publication.

Therefore, I believe it is ready for acceptance and recommend proceeding with the publication of this revised version. No further modifications are necessary at this stage.

Reviewer #4: The contribution of this revised manuscript is clear and good. All comments from the reviewers were very good, and the author’s answer was very good and satisfied. Therefore, my recommendation is to accept the revised manuscript that has Ref. No.: PONE-D-24-36350R1 for publication.

7. PLOS authors have the option to publish the peer review history of their article (what does this mean?). If published, this will include your full peer review and any attached files.

Reviewer #1: No

Reviewer #4: No

---

## [Editor Report · Acceptance letter]

2 Dec 2024

PONE-D-24-36350R1 

PLOS ONE

Dear Dr. Cai, 

I'm pleased to inform you that your manuscript has been deemed suitable for publication in PLOS ONE. Congratulations! Your manuscript is now being handed over to our production team.

Kind regards, 

on behalf of

Dr. Khan Bahadar Khan 

Academic Editor

PLOS ONE